# A Controller-Driven Approach for Opportunistic Networking

MariaCarmen de Toro *, Carlos Borrego and Sergi Robles

Department of Information and Communications Engineering (dEIC), Universitat Autònoma de Barcelona, 08193 Bellaterra, Spain
* Correspondence: mariacarmen.detoro@uab.cat

**Abstract:** Opportunistic networks (OppNets) leverage opportunistic contacts to flow data across an infrastructure-free network. As of yet, OppNets' performance depends on applying the most suitable forwarding strategy based on the OppNet typology. On the other hand, software-defined networking (SDN) is a paradigm for wired networks that decouples the control and data planes. The control plane oversees the network to configure the data plane optimally. Our proposal uses SDN-like controllers to build a partial overview of the opportunistic network. The forwarding strategy uses this context information to achieve better network performance. As a use case of our proposal, in the context of an OppNet quota-based forwarding algorithm, we present a controller-driven architecture to tackle the congestion problem. Particularly, the controller-driven architecture uses the context information on the congestion of the network to dynamically determine the message replication limit used by the forwarding algorithm. A simulation based on real and synthetic mobility traces shows that using context information provided by the controller to configure the forwarding protocol increments the delivery ratio and keeps a good latency average and a low overhead compared with the baseline forwarding protocols based on message replication. These results strengthen the benefits of using supervised context information in the forwarding strategy in OppNets.

**Keywords:** congestion control; data forwarding; intermittently connected networks; opportunistic networks

## 1. Introduction

The Internet is growing fast with ever-increasing machine-to-machine communication (M2M), the expansion of micro and proximity services, the use of cloud services, and the continuous increase in the number of connected devices, among other reasons. The International Telecommunication Union report for the years 2020 to 2030 [1] estimates that there will be 100 billion connected devices by the year 2030. The fastest-growing mobile device category is M2M, followed by smartphones and other smart devices.

Improving the legacy infrastructure to cope with the demand in terms of bandwidth, coverage, quality of service (QoS), and specific requirements for emerging applications is costly and, therefore, not the ultimate solution. Thus, offloading traffic from core networks is a concern. For that matter, device-to-device (D2D) communication [2] performs direct transmissions between peers in range without needing a base station.

A type of network based on D2D communication is opportunistic networks (OppNets). OppNets are characterized by the mobility of their nodes, which leads to an undefined network topology that hinders contemporaneous end-to-end connectivity. Therefore, in OppNets, communication is led by the contact opportunity between peers. This paradigm is very convenient in networks as vehicular ad hoc networks [3], mobile wireless sensor networks [4], pocket-switched networks [5], people-centric networks [6], and mesh networks [7], among others.

OppNets, due to the mobility of their nodes, are prone to frequent disconnections, segmentation, and long delay paths. Hence, traditional routing schemes based on end-to-end connectivity are not applicable. Therefore, OppNets nodes function as routers which

use the store-carry-and-forward (SCF) principle [8] to forward data from the source to the destination on a hop-to-hop basis. Moreover, in OppNets, routing efficiency directly affects the network performance and is highly coupled with the type of application running over the OppNet [9]. In this environment, routing orchestration is a challenge.

In this regard, software-defined networking (SDN) [10] is a network paradigm applied to connected networks that programmatically orchestrates the traffic routing and network configuration by using a program called a controller. The controller has an overview of the whole network and makes routing decisions based on this information. Although SDN protocols are based on TCP and, therefore, are not straightforwardly applicable to OppNets, we apply the concept of having a controller to orchestrate the OppNet traffic routing and agents receiving/sending information to the controller.

Hence, the goal of this proposal consists of using the SDN building blocks to build a context-aware system over an OppNet. This context-aware system leverages context information to dynamically configure the OppNet's forwarding policy parameters aiming to achieve better performance. In the proposed context-aware system, some OppNet's nodes perform as SDN-like controllers (controllers) and the rest as SDN-like agents (nodes). The nodes gather context measurements, specifically device measurements, and send them opportunistically on a hop-to-hop basis to the controllers, which will use this information to dynamically tune the forwarding algorithm parameters. The nodes extend the SCF paradigm to (1) gather device context measurements, (2) aggregate the gathered measurements, (3) forward them to the controllers, and (4) apply the received policies generated by the controllers. In this manuscript, we name this paradigm GAFA. We refer to the OppNet extended with the control system and the GAFA functionalities as *controller-driven* OppNet. In developing this proposal, we make the following contributions:

- Design of the architecture of a novel context-aware system for OppNets inspired by the SDN paradigm where nodes operate based on the GAFA paradigm to feed controllers with device-context information and apply the controllers' policies. The controllers use that information to tune the forwarding algorithm parameters through configuration policies emitted to the nodes to obtain a better network performance.
- Use of the controller-driven OppNet architecture for tackling the congestion in an OppNet characterized by a high unpredictability of the nodes' mobility and a multi-copy replication forwarding strategy. Specifically, the OppNet's controllers orchestrate the value of the replication limit of the forwarding algorithm based on the buffer occupancy readings gathered by the nodes.
- An evaluation of the performance and benefits of the controller-driven OppNet for the use case of congestion control. To evaluate our proposal, we have simulated diverse network scenarios based on real and synthetic mobility traces using different message generation distributions. We have evaluated the controller-driven OppNet on the basis of the standard performance metrics for OppNets. Furthermore, we have run the aforementioned simulations over an OppNet without the control layer (context oblivious) using an epidemic and a quota-based forwarding protocol. We have compared and evaluated the performance of both configurations. We have proven that a controller-driven OppNet performs better than a context-oblivious one.

The rest of the paper is structured as follows. Section 2 introduces the related work in the field of opportunistic networks, focusing on data forwarding and congestion control. Section 3 presents the controller-driven OppNet architecture. Next, in Section 4, the controller-driven OppNet architecture is used to manage congestion. The paper follows with Section 5, where through simulation-based experimentation, we evaluate the performance of the controller-driven OppNet congestion use case, and we compare these results with the performance obtained by a non-controlled OppNet. Finally, Section 6 contains the conclusions drawn from this work.

## 2. Related Work

First, this section describes the current state of the field of opportunistic networks. This proposal's targeted OppNet uses a multi-copy forwarding strategy prone to congestion, so this section accosts congestion control mechanisms for OppNets. Finally, this section develops on the related work in the scope of context-based routing because this proposal's goal consists of using context knowledge to tune the OppNet's multi-copy-based forwarding algorithm.

### 2.1. Opportunistic Networks Overview

An OppNet is a structureless multi-hop network built upon fixed and mobile nodes via wireless links. Due to the mobility of the nodes, OppNets are prone to disruptions, segmentation, and long delay paths. Under these conditions, where there is no guarantee of an end-to-end path between the source and destination at a specific instant in time, the TCP/IP protocol suite is not effective. Hence, the communication in an OppNet is driven by the direct contact opportunity between peers in range, and data flow is achieved by exploiting the pairwise contact opportunity provided by the nodes' mobility. OppNets have been conceived as a complement to connected networks to provide connectivity under specific conditions. Therefore, they are targeted for well-defined practical use case applications characterized by having a limited or even nonexistent infrastructure [8]. Trifunovic et al. [11] stated that the emerging technologies providing global Internet connectivity are not the ultimate solution to settle the classical target OppNet applications domain yet. Moreover, despite the fact that numerous OppNet proposals are formulated as a prototype, there are several commercial solutions [11] and several realistic prototypes, such as [12,13], among others.

### 2.2. Context-Based Routing in Opportunistic Networks

OppNets are prone to long delay paths, disconnections, and segmentation, hindering end-to-end connectivity. Hence, traditional routing based on contemporaneous end-to-end connectivity is not feasible. Therefore, in OppNets, data forwarding is driven hop-to-hop using the store-carry-and-forward (SCF) paradigm originally designed for DTNs [14]. This forwarding paradigm consists of the node *storing* the data and *carrying* the data along the network according to the node's mobility until a contact opportunity occurs and then *forwarding* the data to the contacted node.

Data forwarding is principal in OppNets as application deployment relies on the forwarding as a guarantee of their particular QoS requirements [14]. Seeking forwarding efficiency, Jain et al. [14] state that context information helps to make a more efficient forwarding. CC et al. [15] classified data forwarding algorithms into two main categories depending on the context information used to make routing decisions: social-based routing and pure opportunistic routing. The latter mainly considers device context information.

Under the pure opportunistic routing category, sound forwarding strategies have been proposed in the literature. Those proposals fit well under determined network conditions and application requirements. In this regard, flooding-based strategies, consisting of message replication, have proven to maximize the delivery ratio with a low latency when the OppNet is characterized by the unpredictable nodes' movement [9]. Under the aegis of multi-copy forwarding, the epidemic flooding approach proposed by Vahdat et al. [16] is a context-oblivious strategy prone to suffer from the congestion derived from the replication overhead. Spyropoulos et al. [17] addressed the congestion overhead by establishing a static configured replication quota. Context-aware strategies aim to reduce the effects of a naive replication by calculating the utility of a relay based on historical information. CC et al. [15] highlighted the most relevant routing proposals in this category.

Finally, Boldrini et al. [18] pointed out the relevance of a contextual middleware to manage context information in an OppNet. In this regard, but in the context of wired networks, SDNs [10] decouple the control from the network devices in a control plane where a software named controller gathers network information from the data plane to

build an overview of the network. The controller uses this information to orchestrate the network data flows and resources optimally. To our knowledge, SDN and OppNets have been converged by Li et al. [19]. The authors applied the SDN paradigm in OppNets to implement a mobile crowdsensing system. Nevertheless, southbound communication relies on a cellular network.

*2.3. Congestion Control in OppNets*

In OppNets, traditional mechanisms based on contemporaneous end-to-end connectivity to provide feedback regarding congestion are unsuitable. Furthermore, due to the node's mobility, congestion at a link level is very rare; thus, buffer overload is the main issue. OppNets use the SCF paradigm to forward data across the network. In this paradigm, if a node is affected by congestion, meaning that the buffer is overloaded, the node will need to reallocate, drop queued messages, or reject incoming ones. Either case is highly undesirable as losing messages caused by the node congestion could lead to a delivery failure. Therefore, congestion control is especially critical for OppNets.

Buffer management is a strategy for congestion control applied on the relayed node [20]. Buffer management determines which messages must be dropped in case an incoming one needs to be fitted in. The basic buffer management policies are based on the local's node information such as message priority, lifetime, size, or delivery probability. Krifa et al. [21] stated that basic buffer management policies as drop-tail, drop-head, etc., are suboptimal. They proposed an optimal policy associating a utility function to each queued message, producing a marginal value for a selected optimization metric (delay or delivery ratio). They used statistical learning about encounters to approximate the global knowledge of the network. Pan et al. [22] proposed a mechanism that integrates all the aspects of buffer management.

Another congestion control strategy applied by the sending node is congestion avoidance [20]. Under this category, Goudar et al. [23] stated that basic congestion measures based on buffer management, such as message drops, are not accurate in detecting congestion. They stated that under the inherent characteristics of a mobile OppNet, congestion may occur before buffers are overwhelmed. They proposed an analytical model where the forwarder node calculates the instantaneous forwarding probability of a relay, and they found that this probability decreases dramatically beyond a certain buffer occupancy (buffer occupancy threshold). The node is considered to be congested when it reaches this occupancy threshold.

Furthermore, Lakkakorpi et al. [24] stated that an effective congestion control system should not be based on the network conditions at the time the message was created. Instead, the congestion control system should consider the current network conditions before relaying the message. They proposed a mechanism where each node, upon a contact, shares its buffer availability. Nodes use this information to determine if a relay node has enough buffer resources to custody the message. Thomson et al. [25], measured the congestion as the ratio of drops over message replication per node. They used this information to adjust the replication limit of the messages. Goudar et al. [26] proposed a probabilistic model using an estimator to predict the average buffer occupancy of the nodes in the network. They used this information to discard relay nodes without enough storage to hold the messages to be relayed. Similarly, Batabyal et al. [27] derived a steady-state probability distribution for buffer occupancy.

## 3. Control Layer Architecture

This proposal uses the controller concept from the SDN architecture. We have designed a control layer running on top of the convergence layer of the nodes as a context-aware system. Some of these nodes, the ones selected to be controllers, run the controller module of the control layer. Similar to the SDN controller, the proposed controllers keep an overview of the network by gathering network measurements from the data plane. On the other hand, in OppNets, the control and data planes are coupled in the node. Thereby,

the controller-driven OppNet nodes perform the GAFA functionality introduced in Section 1 for gathering, aggregating, and disseminating network measurements.

Any node could potentially be a controller. Whether a node functions as a controller depends on the nature of the network. In a vehicular network (VANET), the controllers could be the roadside units; in an information-centric network, they could be well-connected nodes. For this particular work, we consider a generic OppNet. Thus, the generic criteria of selecting the more central nodes, i.e., the ones with more contacts, has been applied.

Figure 1 shows a summary of how the control layer implements the node's GAFA and the controller functionalities. Firstly, the nodes in the OppNet sense local context measurements (Figure 1a). Detailed information can be found in Section 3.1. The nodes disseminate an aggregation of context measurements upon a contact (Figure 1b). Section 3.3 develops the aggregation methodology and Section 3.2 shows how the dissemination of context information is performed. Finally, Figure 1c shows how controllers process the received context information to obtain a prediction of a network indicator at a future time (Section 3.5). From this prediction, the controller determines an action to be performed by the nodes consisting of the modification of a forwarding algorithm parameter (Section 3.6). The controller disseminates this action upon a contact with another node (Section 3.7).

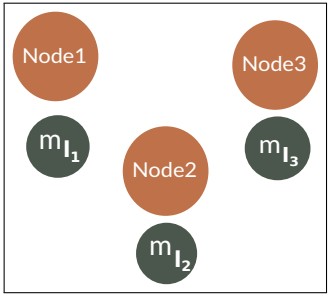

(**a**) Taking network measurements.

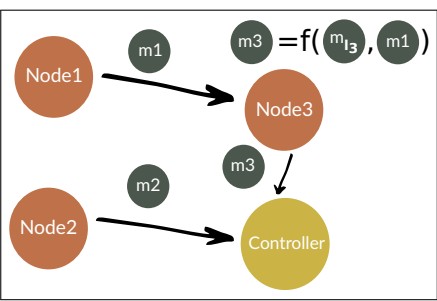

(**b**) Measurements aggregation/dissemination.

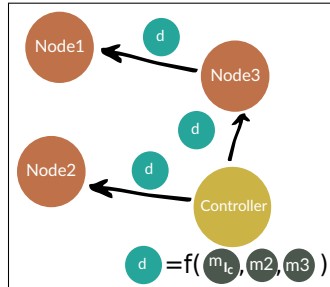

(**c**) Directive creation and dissemination.

**Figure 1.** Node's GAFA functionality: (**a**) gathering measurements, (**b**) aggregating and forwarding them, (**c**) applying the directive created by the controller.

A list of abbreviations is provided in Abbreviations section to help read the rest of the document.

### 3.1. Control Metadata

The control metadata used in the control layer are context measurements and control directives. When two nodes come into range, they disseminate control metadata before delivering or relaying buffered messages. The following two sections describe the aforementioned concepts.

#### 3.1.1. Context Indicator Measurement

A *context indicator measurement* is the local reading of a context indicator taken at a time by the nodes and the controllers (Figure 1a). The controller receives those measurements upon opportunistic contacts with nodes (Figure 1b) and builds an overview of the network based on this information (Figure 1c). In an OppNet, the contact time, bandwidth, and nodes' energy are limited resources, so we have opted to aggregate those measurements (see Section 3.3). Therefore, the context information we consider is this aggregation.

#### 3.1.2. Control Directive

A *control directive* is an action to be performed by the contacted node to modify a forwarding algorithm parameter. A directive is represented as the tuple: $\delta_s = (Id_s, \vartheta_s)$,

where $Id_s$ identifies a forwarding algorithm setting from the list of settings managed by the controller, and $\vartheta_s$ is the value for this setting. The controller generates the directive based on context information (Figure 1c) aiming to improve the forwarding performance. The controller disseminates this directive to the nodes. When a node receives the directive, it applies it by modifying the node's forwarding setting $Id_s$ with the new value $\vartheta_s$. As examples of forwarding settings, we could consider the message TTL, weights, and thresholds intrinsic to the forwarding algorithm, among others.

### 3.2. Context Measurements Dissemination

The node that receives a context measurement stores it in an indexed list. Considering $\{n_1, \ldots, n_z\}$ as the set of nodes in the network at time $t$, the indexed list of received context measurements for the node $n_i$ for $1 < i \leq z$ is represented as $M_i = [m_j \mid 1 \leq j \leq z]$ where $M_i(j) = m_j$.

Figure 2 shows how a node ($n_1$) disseminates its context measurement when comes into range with another node ($n_2$). The control metadata are shared bidirectionally, hence, $n_2$ will follow the same flow when it is its turn to do the dissemination.

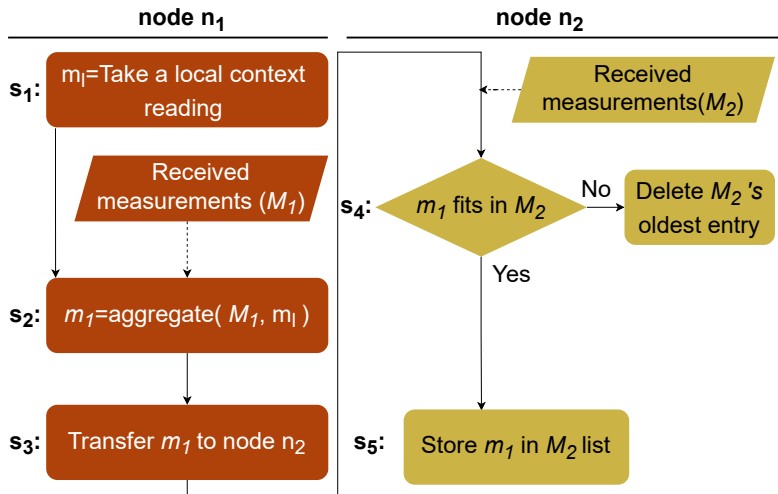

**Figure 2.** Context measurements dissemination when two nodes are in range.

In Figure 2, *step $s_1$*, when node $n_1$ contacts node $n_2$, $n_1$ takes a local context indicator reading, $m_l$. In *step $s_2$*, $n_1$ aggregates this context reading along with all the context measurements received by contacted nodes stored in the list $M_1$. The aggregation process is described in Section 3.3. In *step $s_3$*, the aggregated measurement, $m_1$, is shared with the contacted node $n_2$ which stores it in its context measurements list $M_2$ (*steps $s_4$ and $s_5$*). At this point, $n_2$ follows the same flow to calculate $m_2$ with the slight difference that it will not use the entry $M_2[n_1]$. This entry contains the received measurement $m_1$ built out of the information provided by $n_1$. Using $m_1$ would interfere with the network perception $n_2$ is about to share with $n_1$.

### 3.3. Context Measurements Aggregation

Considering $\{n_1, \ldots, n_z\}$ as the set of nodes in the network at time $t$, we represent the aggregated context measurement generated by node $n_i$ for $1 \leq i \leq z$ as the tuple $m_i = (v_i, \eta_i, t_{c_i})$, where $v_i$ is the aggregated result, $\eta_i$ is the number of measurements used for the calculation of $v_i$, and $t_{c_i}$ is the time when the calculation was made. The node $n_i$'s local context measurement is represented as $m_{l_i} = (v_{l_i}, 1, t_{c_i})$, where $\eta_i = 1$ as $v_{l_i}$ is a straight reading, not the result of an aggregation.

The process that performs the aggregation is *getAggregatedContextMeasurement* (*line 28*) in Algorithm 1. First of all, the node $n_i$ gets the local context reading, $v_l$ (*line 29*). The node uses a two-factor weighted average to aggregate all the context measurements stored in

$M_i$ and its own context reading $v_l$. Each of the context measurements ($m_j$) stored in $M_i$ is weighted by two factors: (1) the number of aggregations used to create it ($\eta_j$)—in the case of the local measurement, this value would be 1—and (2) its decay ($d_j$) at the current time $t$. In the case of the local measurement, the decay would be 1 (no decay).

The following logistic function calculates the decay of a measurement at time $t$:

$$d(t) = \frac{1}{1 + r^2 t} \tag{1}$$

where $r$ is the reduction factor or decay degree to apply to obtain a certain decay. By isolating this variable we obtain:

$$r(t) = \left( \frac{1 - d}{td} \right)^{\frac{1}{2}}. \tag{2}$$

This resulting equation is used to obtain the necessary decay degree ($r$) to be used in (1) to obtain the desired decay at a particular time $t$. The decay is inversely proportional to time ($t$), and it ranges from 0 to 1, where a decay of 1 means no decay. With a decay of 1, the congestion reading is not lowered when aggregated. In contrast, the older the reading is, the higher the decay (lower value), hence the more diminished the reading is when the controller aggregates it along with the other readings received during the aggregation interval.

The aforementioned two-factor weighted average used to aggregate the context measurements in $M_i$ along with the perceived local context reading $v_l$ (*lines 33–41*) is:

$$v_i = \sum_{j=1}^{k} v_j \left( \alpha \frac{\eta_j}{\sum_{p=1}^{k} \eta_p} + (1 - \alpha) \frac{d_j}{\sum_{p=1}^{k} d_p} \right) \tag{3}$$

where $v_i$ is the value resulting from this aggregation; $k$ is the size of $M_i$ plus one (to include the local context reading), $v_j$ is the *value* of the measurement being aggregated ($m_j = M_i[j] = (v_j, \eta_j, t_{c_j})$); $\eta_j$ is the number of aggregations used to generate $m_j$, $d_j$ is the decay of $m_j$ at the current time, and $\alpha$ is the weight factor, specified as a control setting (see Section 5.4.2), used to weigh the two factors of the weighted average. The function *getSumOfNrofAggrs* at *line 17*, calculates the normalizing factor applied in (3) over the number of aggregations the context measurement being processed is formed by ($\sum_{p=1}^{k} \eta_p$). Similarly, the function *getSumOfDecays* (*line 1*) calculates the normalization factor to be applied over the decay of a measurement ($\sum_{p=1}^{k} d_p$).

Finally, the node $n_i$ creates the aggregated measurement $m_i = (v_i, \eta_i, t_{c_i})$ where $v_i$ is the calculated aggregation value, $\eta_i$ is the number of entries in $M_i$ that have been aggregated plus the node's own reading (*lines 32, 39*), and $t_{c_i}$ is the current time. At this point, *step 3* in the flowchart in Figure 2, the calculated aggregated measurement $m_i$ is transferred to the contacted node.

### 3.4. Controller Architecture

The goal of a controller is to have an overview of its nearby part of the network, considering that the mobility nature of the nodes keeps changing the network's topology. This mobility brings on a network "segmentation" in terms of groups of nodes that eventually are connected between them. Ideally, some controllers would be required to cover all the possible network segments.

The controller operates opportunistically, i.e., its actuation is triggered by contacting another node or controller. When that happens, the controller shares both its aggregated context measurement and a directive with the contacted node.

---

**Algorithm 1** Context measurements aggregation algorithm.

---

▷ $M_i$: List of context measurements received from contacted nodes.
▷ *hostID*: Identifier of a host.
▷ *excludeHost*: Id of the host whose measurement in $M_i$ will not be used.
▷ *sumDecays*: Sum of the decays of all the measurements in $M_i$.
▷ $M_i\_keys$: Measurement list indexes.
▷ *m*: A received context measurement represented by the tuple $(v, \eta, t)$
▷ *threshold*: Decay threshold under which the measurement is considered to be expired.
▷ *sumNrAggr*: Sum of the # aggregations a measurement is made of.
▷ *#aggrEntries*: # of elements in $M_i$ that have been aggregated.
▷ *aContextReading*: A local context measurement.
▷ $v_i$: Aggregated measurement value.

1: **function** GETSUMOFDECAYS($M_i$,excludeHost)
2: 　　*sumDecays* ← 0
3: 　　**for all** *hostID* ∈ $M_i\_keys$ **do**
4: 　　　　**if** *hostID* != *excludeHost* **then**
5: 　　　　　　*m* ← $M_i$[*hostID*]
6: 　　　　　　*d* ← *decay*(*current_time* − *m*[*t*])
7: 　　　　　　**if** *d* < *threshold* **then**
8: 　　　　　　　　$M_i$.remove(hostID)
9: 　　　　　　**else**
10: 　　　　　　　　*sumDecays* += *d*
11: 　　　　　　**end if**
12: 　　　　**end if**
13: 　　**end for**
14: 　　*sumDecays* ++　　　　　　　　　　▷ Adding the decay of my own reading.
15: 　　**return** *sumDecays*
16: **end function**
17: **function** GETSUMOFNROFAGGRS($M_i$, excludeHost)
18: 　　*sumNrAggr* ← 0
19: 　　**for all** *hostID* ∈ $M_i\_keys$ **do**
20: 　　　　**if** *hostID* != *excludeHost* **then**
21: 　　　　　　*m* ← $M_i$[*hostID*]
22: 　　　　　　*sumNrAggr* += *m*[$\eta$]
23: 　　　　**end if**
24: 　　**end for**
25: 　　*sumNrAggr* ++　　　　　　　　　　▷ Considering its own reading.
26: 　　**return** *sumNrAggr*
27: **end function**
28: **function** GETAGGREGATEDCONTEXTMEASUREMENT($M_i$, excludeHost)
29: 　　$v_l$ ← *aContextReading*
30: 　　*decays* ← *getSumOfDecays*($M_i$, *excludeHost*)
31: 　　*aggrs* ← *getSumOfNrofAggrs*($M_i$, *excludeHost*)
32: 　　*#aggrEntries* ← 1　　　　　　　　▷ Considering its own reading.
33: 　　$v_i \leftarrow v_l((\alpha \frac{1}{aggrs}) + ((1-\alpha)\frac{1}{decays}))$
34: 　　**for all** *hostID* ∈ $M_i\_keys$ **do**
35: 　　　　**if** *hostID* != *excludeHost* **then**
36: 　　　　　　*m* ← $M_i$[*hostID*]
37: 　　　　　　*d* ← *decay*(*current_time* − *m*[*t*])
38: 　　　　　　$v_i \mathrel{+}= m[v]((\alpha \frac{m[\eta]}{aggrs}) + ((1-\alpha)\frac{d}{decays}))$
39: 　　　　　　*#aggrEntries* ++
40: 　　　　**end if**
41: 　　**end for**
42: 　　**return** $m_i = (v_i, \#aggrEntries, t)$
43: **end function**

---

To generate a directive, the controller implements the closed-loop control system, also known as feedback control system [28], showed in Figure 3. A closed-loop control system is a control system that maintains a constant relation between the output of the system ($\mathfrak{c}$: controlled variable) and the desired value ($\mathfrak{r}$: the reference input) by subtracting one from the other as a measure of control.

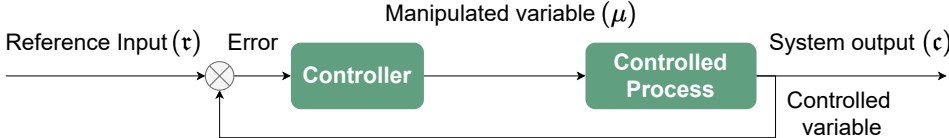

**Figure 3.** Closed-loop control system for congestion control.

In the proposed control system, the controller feeds from incoming aggregated context measurements ($m_i$) and determines the value of the manipulated variable $\mu$ out of those measurements and the system's reference input $\mathfrak{r}$. The resulting manipulated variable $\mu$ is encapsulated in an outgoing directive and shared with any node that might come into contact.

Additionally, Goudar et al. [26] stated that considering the current condition of the OppNet to trigger an actuation is not effective as it would be too late, given the variability of the network. Following their considerations, proactively, we decide the actions to be taken based on predicting the network situation. By this time, presumably, the directive would have been propagated and applied throughout the network. Indeed, if the control system would directly inject the raw aggregated context measurements received from other nodes to the controller for generating a directive containing $\mu$, when eventually this directive would reach the nodes, the network situation might have changed and, possibly, the directive would be no longer adequate for the current situation. The following section shows how we predict the value of a context indicator.

### 3.5. Context Indicator Prediction

The controller anticipates the context indicator value using a two-step strategy. Firstly, instead of directly considering the received context measurements, the controller aggregates them for a configurable time ($\hat{t}$) using (3). This step results in two lists: $\check{M}$ and $\check{T}$. $\check{M}$ contains the aggregation of the measurements received for periods of $\hat{t}$ seconds. This list is represented as $\check{M} = [\check{m}_j \mid 1 \leq j \leq \check{z}]$, where $\check{z}$ is the max size of $\check{M}$. $\check{T}$ contains the time when each entry in $\check{M}$ was calculated and is represented as $\check{T} = [\check{t}_j \mid 1 \leq j \leq \check{z}]$. Secondly, once several aggregated samples of context measurements are available, these are used as an input of a linear regression ($\rho$) to calculate the prediction of this context indicator value for time $t_{t+n}$:

$$m_{t+n} = \rho(t_{t+n}, \check{M}, \check{T}) \tag{4}$$

where $\check{M}$ and $\check{T}$ are sliding lists of size $\check{z}$, $t_{t+n}$ is the time ahead for the context indicator prediction and it is calculated as $t = t + n$ where $n$ is an offset, and $m_{t+n}$ is the resulting predicted value of the context indicator value at time $t_{t+n}$.

Algorithm 2 shows how the controller calculates the prediction of a context indicator. The *addContextMeasurement* procedure is executed by the controller when a contacted node shares its aggregated context measurement ($m$). This measurement is stored in the controller's received measurements list $M_i$ (*line 2*). If the window time for receiving measurements from contacted nodes has expired (*line 3*), the controller aggregates all the received aggregated context measurements in the list $M_i$ and its local context measurement, by using the procedure *getAggregatedContextMeasurement* (*line 5*) defined in Algorithm 1. This aggregation result ($\check{m}$) is stored in the list $\check{M}$ along with the current time (*lines 6–7* of Algorithm 2). The controller uses the entries in the above lists to calculate a prediction of the value of the context indicator ($m_{t+m}$) using a linear regression function $\rho$ (*line 10*). The controller maintains the size of the aggregation calculations list $\check{M}$ at the constant

value $\check{z}$ by using a FIFO discarding policy (*lines 12–13*). Notice that at least two inputs are needed to use the linear regression function $\rho$ (*lines 8* and *9*). If this condition is not fulfilled, the predicted measurement ($m_{t+n}$) assumes the value of the $\check{m}$ calculated at *line 5* (*line 15*). Once $m_{t+n}$ is calculated, $M_i$ is emptied, ready to receive new context measurements from other contacts (*line 19*). A new aggregation window period (*aggrTimeout*) is configured, so all the measurement gathering and the prediction process starts over (*line 20*).

---

**Algorithm 2** Controller's context indicator prediction.

---

    ▷ $M_i$: List of context measurements received from contacted nodes.
    ▷ $m$: Received context measurement (controlled variable).
    ▷ $t$: Current time.
    ▷ *aggrTimeout*: Timeout for aggregating incoming context measurements.
    ▷ $\hat{t}$: Time period for aggregating incoming context measurements.
    ▷ $\check{m}$: Aggregation of the received context measurements during $\hat{t}$ s.
    ▷ $\check{M}$: Sliding list of the aggregations so far.
    ▷ $\check{T}$: Sliding list of the times when the aggregations were performed.
    ▷ $t_{t+n}$: Prediction time.
    ▷ $m_{t+n}$ Predicted value of the context indicator at $t_{t+n}$.
    ▷ *directive*: Manipulated variable ($\mu$) encapsulated in a directive

1:  **function** ADDCONTEXTMEASUREMENT($m$)
2:     $M_i.add(m)$
3:     **if** $t >= aggrTimeout$ **then**
4:         //The window time for receiving context measurements has finished.
5:         $\check{m} \leftarrow getAggregatedContextMeasurement(M_i, NULL)$
6:         $\check{M}.add(\check{m})$
7:         $\check{T}.add(t)$
8:         **if** $\check{M}.size() > 0$ **then**
9:             **if** $\check{M}.size() > 1$ **then**
10:               $m_{t+n} = \rho(t_{t+n}, \check{M}, \check{T})$
11:               //Just keep $\check{z}$ values.
12:               $\check{M}.removeEldestN()$   ▷ Sliding the list.
13:               $\check{T}.removeEldestN()$
14:             **else**
15:               $m_{t+n} = \check{m}$
16:             **end if**
17:             $directive \leftarrow createDirective(m_{t+n})$
18:         **end if**
19:         $M_i.clear()$
20:         $aggrTimeout \leftarrow t + \hat{t}$
21:     **end if**
22: **end function**

---

### 3.6. Directive Generation

The function *createDirective*($m_{t+n}$) (*line 17*, Algorithm 2) is called to generate a directive encapsulating the context indicator prediction ($m_{t+n}$). The former function, defined in Algorithm 3, calculates the manipulated variable's value ($\mu'$) from of the context indicator prediction and the reference value (*line 4*). Next, $\mu'$ is encapsulated in a directive: $\delta_s = (Id_s, \mu')$ (*line 6*).

As previously mentioned, the reception of a context measurement after contacting a node triggers the generation of a directive. Nevertheless, if the time window for receiving context measurements is set to a high value, it would take a controller a long time to generate a directive. Hence, the nodes would not receive any directive to adjust their initial configured manipulated variable ($\mu$) based on the current network condition, and they would have the perception that there is no controller nearby. Therefore, to prevent this situation, the controller is configured to generate a directive periodically, provided no directive

has been generated opportunistically during this period. This directive encapsulates the last calculated manipulated variable ($\mu$), and it acts as a beacon announcing the presence of a nearby controller. Algorithm 3 describes the above behaviour.

---

**Algorithm 3** Directive creation algorithm.

  $\triangleright$ $m_{t+n}$: Predicted measurement.
  $\triangleright$ *opp*: Execution mode's flag.
  $\triangleright$ $\mu$: Manipulated variable's current value.
  $\triangleright$ $\mu'$: Manipulated variable's new value.
  $\triangleright$ $\mathfrak{r}$ Controller system's reference input.
 1: **function** CREATEDIRECTIVE($m_{t+n}, opp =$ FALSE)
 2:  $\mu' = \mu$
 3:  **if** $opp == false$ **then**
 4:   $\mu' \leftarrow$ apply controller_adjustment($\mathfrak{r}, m_{t+n}$)
 5:  **end if**
 6:  return new Directive($\mu'$)
 7: **end function**

---

Notice that the function *createDirective* in Algorithm 3, receives the parameter *opp* which indicates whether the function is executed periodically, as described above or opportunistically after receiving a context measurement from a contacted node (see Algorithm 2). In the case of a periodic execution of the function *createDirective*, the new value for the manipulated variable ($\mu'$) is directly the current one ($\mu$) (*lines 2* and *6* in Algorithm 3). In the case of an opportunistic execution (*line 3*), the controller calculates the new value for the manipulated variable (*line 4*).

### 3.7. Directive Dissemination

Although it is a controller that generates a directive, it is stored, carried, and forwarded by any node in the network that receives it. Figure 4 shows this behaviour. When a node receives a directive ($dir_{n_1}$) (*step $s_4$*), if the directive is newer than the one the node might be carrying (*steps $s_5$–$s_6$*), the node discards the old one (*step $s_7$*), executes the action encapsulated in the new directive (*step $s_8$*), and stores, carries, and forwards the new directive (*step $s_9$*).

Discarding the older directive is a measure to deal with possible inconsistent directives as several controllers are allowed in the network. The newer a directive is, the closer the node is to the controller and, therefore, the more appropriate the directive is. If a node receives several directives, if they are consistent, it means that the controllers are also nearby and are mainly sensing the same context. Conversely, if the received directives are inconsistent, it indicates that the controllers are sensing different parts of the network. In this case, the node likely belongs to the network "segment" controlled by the controller generating the newer directive.

In the following section, we will describe how to apply the control layer for the specific use case of congestion control.

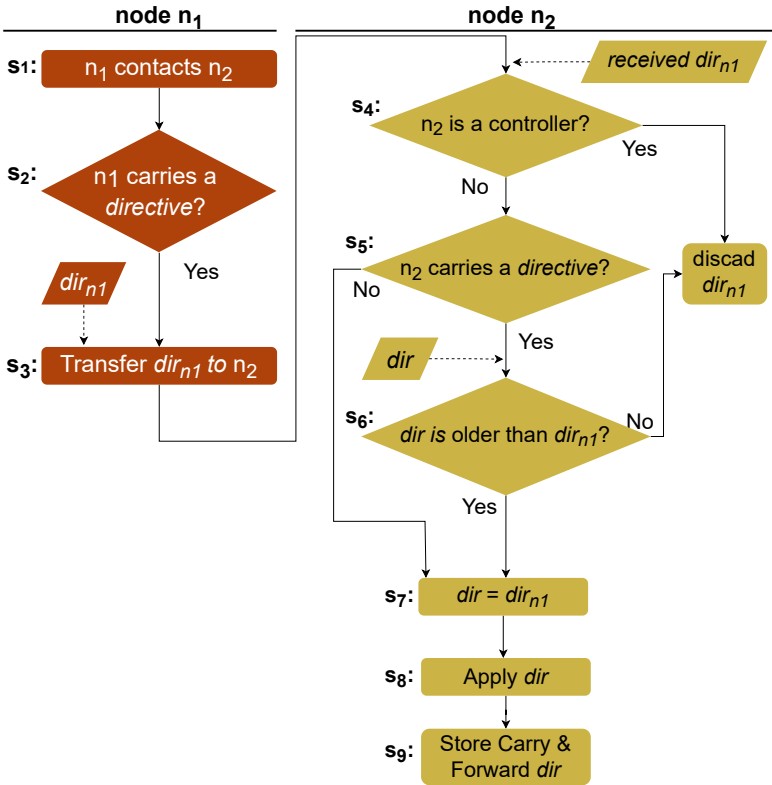

**Figure 4.** Disseminating a directive when two nodes are in range.

## 4. Use of the Controller-Driven OppNet Architecture to Manage Congestion

This section describes how the control layer, presented in Section 3, is used over an OppNet characterized by a multi-copy forwarding strategy to manage the congestion intrinsic to replication [20]. Specifically, the controller-driven OppNet will approximate how congested the network is and, out of this knowledge, will adjust the replication limit of the newly created messages and the ones being forwarded.

In the following sections, we specialize for the use case of congestion control: (1) the data message representation the control layer will work with, (2) the adaptation of the controller architecture (presented in Section 3.4) for this particular use case, (3) the context indicator prediction (introduced in Section 3.5), and (4) the directive generation (Section 3.6) and the directive dissemination (Section 3.7). Finally, we summarize the buffer management techniques used by the control layer as a congestion control strategy.

### 4.1. Control Layer Data Message

In the scope of forwarding algorithms based on message replication, one of the forwarding algorithm parameters is the replication limit of the message. The replication limit determines the total number of copies of the message allowed to exist in the network. As seen in Section 2.2, multiple replication strategies exist. For this particular use case, we consider a forwarding algorithm that uses a binary replication scheme consisting in relaying a copy of the message to the contacted node and reducing by half the replication limit of both the node's message and the relayed copy of it.

The control layer encapsulates the data messages generated by the application layer in the tuple $g = (\mathfrak{a}, l, \varrho, \varphi)$. From this tuple, $\mathfrak{a}$ is the data message generated at the application layer. $l$ is the replication limit of the message (in the case of a binary replication scheme, the message can be relayed $\lceil l/2 \rceil$ times). $\varrho$ is the number of times this particularly message copy has been relayed. This field is incremented each time the message is relayed to the next hop. $\varphi$ is the *alive* flag. This flag is set to false to indicate that this message is marked to be deleted in case buffer space is required. This message is not deleted straightforwardly

as it could happen that the next contact could be the message destination. Furthermore, it could occur that applying another directive would update this message's replication limit to a value equal to or higher than one, providing the message with more chances to be delivered. With this congestion measure, the messages with the field $\varphi$ set to *false* are reactively removed in case of need, but also, in a proactive way, the message is given a chance to be carried along the network while there is no need for buffer space. This strategy requires the node to not consider the messages with the $\varphi$ field set to *false* when calculating its buffer occupancy measurement.

### 4.2. Control Layer Tailored for Congestion Control

The control layer proposed in Section 3.4 has been adapted to use a congestion policy to efficiently limit the number of messages in the network to limit congestion. This congestion policy dynamically determines the replication limit for a newly created message based on the network congestion perception of a controller.

As a network congestion measure, we use the node's buffer occupancy rate ($o$):

$$o = \frac{\sum_{i=1}^{n} sizeof(g_i)}{b} \tag{5}$$

where $n$ is the number of buffered data messages, *sizeof* is the function that returns a message size in bytes, $g_i$ is a buffered message, and $b$ is the buffer capacity in bytes.

The generic closed-loop control system in Figure 3, for the use case of congestion control, is specialized in Figure 5 with the slight difference that the controller assumes calculating the difference between the reference input ($\mathfrak{r}$) and the controlled variable ($\mathfrak{c}$). The controller's reference input ($\mathfrak{r}$) is an optimal buffer occupancy congestion interval. This interval consists of a range of buffer occupancy rates, defined by a lower and upper bound, $o_{min}$ and $o_{max}$, respectively. The buffer occupancy is considered optimal when the buffer is neither underused nor close to its maximum capacity.

The controller feeds from the buffer occupancy measurements ($o$) received from contacted nodes and uses those measurements to calculate a buffer occupancy prediction ($o_{t+n}$) following Algorithm 2.

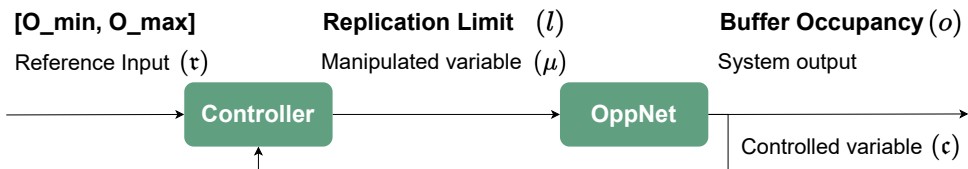

**Figure 5.** Closed-loop control system for congestion control.

### 4.3. Network Congestion Prediction

The controller uses the optimal congestion interval along with the calculated $o_{t+n}$ to determine its prediction of the network's congestion. The network's congestion status falls into three states: *UNDER_USED*, *OPTIMAL*, and *CONGESTED*. The controller uses the following function to sentence which is its prediction of the network's congestion state ($c_{t+n}$):

$$c_{t+n}(o_{t+n}) = \begin{cases} \text{UNDER\_USED} & \text{if } o_{t+n} < o_{min} \\ \text{OPTIMAL} & \text{if } o_{min} \leq o_{t+n} < o_{max} \\ \text{CONGESTED} & \text{if } o_{t+n} \geq o_{max}. \end{cases} \tag{6}$$

### 4.4. Directive Generation for Congestion Management

Based on its congestion prediction ($c_{t+n}$), the controller calculates the maximum number of copies of a message allowed to be in the network ($l$). For that matter, this proposal considers a proportional controller (P-controller) [29] using an additive increase and multiplicative decrease (AIMD) factor to adjust the manipulated variable ($l$). The considered

AIMD factor uses $k_1$ as a multiplicative decrease factor and $k_2$ as an additive increase factor. Out of the predicted network congestion, the P-controller calculates the new replication limit $l'$.

$$l'(c_{t+n}) = \begin{cases} l & \text{if } c_{t+n} = \text{OPTIMAL} \\ l \cdot k_1 & \text{if } c_{t+n} = \text{CONGESTED} \\ l + k_2 & \text{if } c_{t+n} = \text{UNDER\_USED} \end{cases} \tag{7}$$

With this control function, if the predicted network's congestion state is *OPTIMAL*, it is not necessary neither to increase nor decrease the replication limit of a message. When the predicted network's congestion state is *CONGESTED* or *UNDER_USED*, the controller decreases or increases the message's replication limit, respectively.

The controller encapsulates the calculated $l'$ in a directive: $\delta_l = (\mathit{Id}_l, l')$, where $\mathit{Id}_l$ is the identifier of the message replication limit setting and $l'$ is the calculated replication limit. Following the flowchart described in Figure 4, when the controller contacts a node, it forwards the former directive along with the calculation of its buffer occupancy.

### 4.5. Applying a Directive

Applying a directive $\delta_l = (\mathit{Id}_l, l')$ entails two actions: (1) using the encapsulated replication limit $l'$ when creating new data messages and (2) updating all the buffered messages according to the new $l'$. The first action consists of using the new replication limit $l'$ when an application creates a new message ($g = (\mathfrak{a}, l', \varrho, \varphi)$). The second action consists of updating the message's field $l$ of the buffered messages, considering that some are already copies.

Indeed, it would not be accurate to update the buffered messages with the new replication limit $l'$ received through $\delta_l$, as some of these messages are already copies. Therefore, the number of times this message has been relayed (message's $\varrho$ field) is considered. This information is used to calculate the current replication limit of the buffered message, considering that: (1) the message was created with the replication limit set by the last received $\delta_l$ and (2) $\lceil l/2 \rceil$ copies of the message are forwarded at each encounter:

$$l'' = \frac{l'}{2^\varrho} \tag{8}$$

where $l''$ is the remaining replication limit after relaying the message $\varrho$ times.

Algorithm 4 shows the procedure for modifying the replication limit of the buffered messages. For each of the buffered messages (*line 2*), provided the replication limit it was assigned when created was $l'$ (from the received directive $\delta_l = (\mathit{Id}_l, l')$) and taking into account the number of times that the message has been relayed ($\varrho$), the remaining replication limit ($l''$) is calculated using (8) (*line 3*). In case the calculated $l''$ is less than one, it would mean that by starting with the replication limit specified in the received directive, the message would not have any copies left to disseminate at this point and, therefore, the message would not reach the current node. If this is the case, the message's field $\varphi$ is set to *false* (*line 5*), indicating that this message is marked to be deleted in case buffer resources are required.

### 4.6. Buffer Management

The control layer applies a congestion control mechanism based on buffer management. Besides the congestion policy limiting the number of message copies in the network, the control layer applies a hybrid drop policy.

The control layer foresees the network congestion (Section 4.3). Based on the predicted congestion, it limits the number of message copies (Section 4.4) and updates the copies left of the queued messages (Section 4.5). Proactively, if this update results in the message having no copies remaining, the message is marked to be deleted by setting the message's flag $\varphi$ to false. Reactively, in case of buffer overflow, the messages marked to be deleted are dropped. The control layer applies basic drop policies if more buffer space is required. It

first applies a *drop-oldest* policy based on the message's remaining TTL. Next, if necessary, it applies a *drop-head* policy removing the oldest ones.

---

**Algorithm 4** Procedure to update the buffered messages with the $l$ set by a directive.

$\qquad \triangleright \delta_l$, Received directive ($\delta_l = (Id_l, l')$)
$\qquad \triangleright l'$: The replication limit in the received directive ($\delta_l = (Id_l, l')$).
$\qquad \triangleright g$: A message ($g = (\mathfrak{a}, l, \varrho, \varphi)$).
$\qquad \triangleright B$: A buffer to store messages.
1: **function** APPLYDIRECTIVETOBUFFEREDMESSAGES($\delta_l$)
2: $\qquad$ **for all** $g \in B$ **do**
3: $\qquad\qquad l'' = \frac{l'}{2^\varrho}$
4: $\qquad\qquad g[l] \leftarrow l''$ $\qquad\qquad\qquad\qquad$ $\triangleright$ Setting the new rep. limit in the message's field $l$
5: $\qquad\qquad g[\varphi] \leftarrow (l'' < 1) \,?\, false : true$
6: $\qquad$ **end for**
7: **end function**

---

## 5. Experimentation

The controller-driven OppNet, which uses a quota-based multi-copy forwarding with a dynamic message replication limit, is named the control configuration (Control). The control configuration is compared with two *No-Control* multi-copy baseline forwarding algorithms: epidemic (EP) and a quota-based algorithm (Static), both presented in Section 2.2. In epidemic routing, nodes forward messages to every encountered node to achieve maximum network coverage. Static routing sets a static upper bound of the number of message replicas in the network. It distributes half of these copies to each contact (provided the contact does not carry copies of the message yet) until the node has only one copy left, which will carry up to the destination. Both no-control approaches are multi-copy baseline forwarding algorithms considered for benchmarking in opportunistic networking research [30].

This section describes how the control configuration performs. The experimentation methodology follows the guidelines pointed out by Dede et al. [31], and Kuppusamy et al. [32]: (1) appropriate mobility models to favour different congestion degree situations are designed; (2) our proposal is compared with benchmark multi-copy forwarding algorithms; (3) several performance metrics, listed in Section 5.1, are evaluated over the Control and No-Control configurations; moreover, the network performance is also evaluated for the Control configuration for different values of its configuration settings listed in Section 5.4.2; (4) the experimentation setup is detailed and well documented to be reproduced for benchmark purposes; and (5) the simulator provides the link model and the physical aspects are not considered.

### 5.1. Performance Metrics

We use the standard metrics to measure the performance of the Control and No-Control configurations [33]:

**Delivery Ratio ($\sigma$):** measures the ratio of created messages that are delivered to the final destination:

$$\sigma = \frac{\#g_d}{\#g_c} \qquad (9)$$

where $\#g_d$ is the number of delivered messages, and $\#g_c$ is the number of created messages.

**Latency average ($\overline{\lambda}$):** the average time it takes for the created messages to get delivered to their final destination:

$$\overline{\lambda} = \frac{\sum_{i=1}^{w} \lambda_i}{w} \qquad (10)$$

where $w$ is the number of messages delivered to the destination, and $\lambda_i$ is the elapsed time from the message creation to its delivery.

**Overhead ratio ($\theta$):** measures the average of the message copies needed to deliver the message to its final destination:

$$\theta = \frac{\#g_r - \#g_d}{\#g_d} \tag{11}$$

where $\#g_r$ is the number of relayed copies, and $\#g_d$ is the number of delivered messages.

### 5.2. Scenarios

We use four scenarios which use different mobility patterns representing different network conditions. These scenarios are classified into two groups: (1) the scenarios based on real-world mobility traces, available at the Crawdad database [34] and (2) the synthetic scenarios generated by a random waypoint model (RWP) in a grid with reflective barriers where the nodes move at a configured speed for a configured distance. The nodes keep changing direction each time they cover that distance. The scenarios based on real mobility traces are very convenient for evaluating this proposal under real network conditions. In contrast, the synthetic scenarios help us to recreate particular network conditions as emergencies, not yet covered with real mobility traces samples. The considered scenarios are:

**Taxis:** tracks 304 Yellow Cab taxis in the San Francisco Bay area for one week. The traces are available at [35].

**Info5:** tracks the movement activity of 41 students attending the Infocom conference in 2005 over three days [36].

**Campus:** a synthetic map-based scenario that simulates the mobility activity of 80 students at Autonomous University of Barcelona campus, covering an area of 4.5 km $\times$ 3.4 km with defined points of interest (POI) corresponding to eight faculties and the railway station. The students walk throughout the campus arriving/leaving their faculty and the railway station with a certain probability.

**Emergency:** a synthetic RWP-based scenario with 100 nodes randomly walking in an area of one square kilometre. In this scenario, the pattern of message generation changes abruptly in the second half of the simulation, aiming to roughly simulate network conditions under the disrupting events of an emergency.

Each scenario has different node densities. A sparse nature in a scenario entails fewer contact opportunities among the nodes, whereas dense natures are prone to more contacts between nodes. The Taxis scenario is a sparse scenario, and the Campus is a sparse scenario with POI promoting a temporary high density of nodes. Info5 and Emergency are both dense scenarios.

### 5.3. Message Generation Distribution

Depending on the scenario, for the data message generation, we will be using either a constant bit rate (CBR) distribution or an inverted smoothed top hat distribution (ISTH) [37]. Next, we formalize the ISTH for the specific case of a 24 h working day.

#### 5.3.1. Inverted Smoothed Top Hat Distribution (ISTH)

We aim to mimic the network traffic in a working day with the ISTH distribution. The flat region (FR) of the ISTH represents the working hours where the network traffic is the heaviest, i.e., messages will be generated at a higher rate. The transient regions (TR) are shaped in two ways: (1) to logarithmically increase the rate of message generation up to reach the peak rate of the FR; and (2) to logarithmically decrease the rate of the message creation from the peak rate in the FR to a shallow rate when approaching the 24th hour of the day.

The ISTH function is built as a composition of a descendant logistic functions, an ascendant logistic function, and a linear function. Both logistic functions determine the message generation frequency for a time:

- Descendant logistic function (exponential growth rate$(k) > 0$):

$$f(x) = \frac{L_2}{1 + ae^{k(x-x_0)}} \tag{12}$$

- Ascendant logistic function ($k < 0$):

$$g(x) = \frac{L_2}{1 + ae^{-k(x-x_0)}}. \tag{13}$$

These functions are bounded by two limits: $L_2$ and $L_1$. $L_2$ corresponds to the lowest message generation rate (highest value) used to generate very low traffic. $L_1$ corresponds to the highest message generation rate (lowest value) used to generate the highest network traffic. $x - x_0$ is the flexible horizontal translation. No horizontal translation is considered: $x_0 = 0$. $k$ is the exponential growth rate.

From 00:00 a.m. to 09:00 a.m., the descendant function defined in (12) needs to descend from $L_2$ down to $L_1$, i.e., $f(0) = L_2$ and $f(9) = L_1$. Both $L_2$ and $L_1$ limits are specified through the control configuration settings depending on the scenario. Hence, from (12), the only unknown variable ($a$) is isolated:

$$a = \frac{(L_2 - L_1)}{L_1 e^{k9}}.$$

The ascendant logistic function in (13) follows the same process to ascend from $L_1$ at the end of the working hours of the day (17:00 h) up to $L_2$ at (00:00 h): $g(17) = L_1$ and $g(0) = L_2$.

Finally, to build the ISTH function it is necessary to combine the descendant and ascendant logistic functions with a flat linear function that covers the eight working hours of the day (from 09:00 a.m. to 17:00 p.m.). During this time window, messages are generated at $L_1$ rate:

$$h(x) = \begin{cases} 0 \leq x < 9: & f(x) \\ 9 \leq x < 17: & L1 \\ 17 \leq x < 24: & g(x) \end{cases}. \tag{14}$$

### 5.3.2. Scenarios' Message Generation Distribution

Table 1 summarizes the message creation distribution for the different scenarios. The Taxis scenario uses an ISTH distribution that replicates two working days, where the messages are generated each 10 to 60 s uniformly distributed for each working day during eight peak hours. With all the scenario specifics, it is considered a low- to medium-congestion scenario.

The Info5 scenario uses the aforementioned message generation distribution. Given the message generation distribution and the fact that the nodes congregate around the events of the congress, it is considered a medium- to high-congestion scenario.

The Campus scenario uses an ISTH distribution resembling a workday where, during the peak hours, messages are created every 10 s. Hence, this scenario is considered a high-congestion scenario.

For the Emergency scenario, during the first eight hours, messages are generated at the high rate of each 10 s using a CBR distribution followed by eight hours of low-rate message generation (every 80 s). Therefore, Emergency is a variable-congestion scenario.

**Table 1.** Message generation distribution per scenario.

| Settings | Taxis | Info5 | Campus | Emergency |
|---|---|---|---|---|
| Simulation time | 69 h | 70 h | 34 h | 23 h |
| Message distribution | ISTH | ISTH | ISTH | CBR |
| Message generation frequency | FR: [10–60] s; TR: 1800 s | FR: [10–60] s; TR: 900 s | FR: 10 s; TR: 1800 s | 0–8 h: 10 s; 8–16 h: 80 s |
| Congestion level | Low-Medium | Medium-High | High | Medium-High |

### 5.4. Environment Setup

For the experimentation, the Opportunistic Network Environment (ONE) simulator [38] was used, which is designed specifically to simulate OppNets. Recent works show that it is the most used simulator for OppNets [32]. The ONE has proven easy to configure and provides an extensive set of mobility, traffic models, and propagation protocols [31]. The control layer has been developed on top of the simulator's network layer and is available through a public repository (https://github.com/MCarmen/the-one/tree/control, accessed on 3 December 2022).

The simulations over the synthetic scenarios use a traces file with node contacts generated with the built-in RWP model of the simulator to preserve the same scenario over different simulation rounds. Next, we will describe the configuration settings common to all the scenarios and the specific settings by scenario.

#### 5.4.1. Common Configuration Settings

Table 2 lists the common simulation configuration settings. For all the scenarios, the nodes are configured with a WiFi interface with a transmission speed of 100 Mbps and a transmission range of 60 m as an approximation of the WiFi 5 (802.11ac) standard.

In each simulation cycle, any random node in the network creates a message to a randomly selected node to approximate a real-world communication model. For the simulations, it is considered that when two nodes are in range, they have enough time to exchange the control protocol data, the messages to be delivered to the contacted node, and the messages to be relayed.

**Table 2.** Summary of the common simulation settings for all the scenarios.

| Setting | Value | Setting | Value |
|---|---|---|---|
| Network interface | Wi-Fi | Battery | none |
| Transmission speed | 100 Mbps | Type of nodes | pedestrians |
| Transmission range | 60 m | User behaviour | none |
| Interference | none | Application | Single destination |
| Power consumption | none | | |

#### 5.4.2. Scenario and Control Configuration Settings

The nodes' buffer size, the messages' TTL, the messages' size, and message generation frequency, which directly affect the network congestion, vary for each scenario to create different congestion conditions. The first part of Table 3 lists the values for the above scenarios' settings. In this table, the intervals specifying the value for the settings: message generation frequency, message size and walk speed denote a uniform distribution between the two interval limits. Notice that for all the scenarios, the nodes' buffer size is set to 10 M to favour congested situations, mainly when messages are generated at a high rate.

Also, the control layer can be customized through the settings listed in the second part of Table 3. Next, we describe the customisable control settings:

**Number of controllers:** indicates the number of nodes that will act as controllers in the network.

**Optimal congestion interval ($o_{min} - o_{max}$):** as presented in Section 4.2, this setting specifies the optimal range of the node's buffer occupancy.

**Additive increase ($k_2$) and multiplicative decrease ($k_1$) (AIMD):** denote the factor to be added to ($k_2$) and the factor to multiply by ($k_1$) the current replication limit ($l$) to update the former replication limit based on the network congestion status through (7).

**Aggregation interval ($\hat{t}$):** time while the controller gathers congestion measurements (see Section 3.5).

**LR nrof inputs ($\check{z}$):** maximum size of the congestion readings aggregation list $\check{M}$. This list is used as an input for the congestion prediction function. $\check{M}$ works as a sliding list of size $\check{z}$ to consider a recent history of readings for the prediction.

**Prediction time factor ($\phi$):** the multiplicative factor applied over the aggregation interval setting ($\hat{t}$) to determine the time ($t_{n+t}$) for a congestion prediction (Section 3.5, Algorithm 2, *line 10*). $\phi$ is calculated by the equation:

$$t_{t+n} = t + \hat{t}\phi \tag{15}$$

where $t$ is the current time, and $\hat{t}$ is the time interval for aggregating congestion readings.

**Directive generation frequency:** determines the periodicity of the automatic directive generation triggered in case the controller does not receive any congestion reading for this period of time (see Section 3.6).

**Reduction factor for a specific decay ($r$):** the reduction factor to apply to get a certain decay. Used in (1) (Section 3.3).

**Decay threshold:** when a controller receives a congestion measurement with a decay lower than *decay threshold*, it is discarded, and hence, it is not used to estimate the congestion.

**Number of aggregations weight ($\alpha$):** weight factor applied over the number of aggregations a congestion measurement is built on. Used in (3) (Section 3.3).

*5.5. Results*

This section shows and evaluates our proposal (Control) and the No-Control configurations, introduced in Section 5, for different scenarios, in terms of (1) the buffer occupancy, (2) the performance metrics listed in Section 5.1, and (3) the delivery ratio for different values of the controller settings listed in Section 5.4.2. Before delving into the comparison between the Control and No-Control configurations, for the Control configurations, we analyze the replication limit ($l$) it tends toward. Along this section, the Control configuration will be compared to the Static configurations with the same $l$ that the Control tends toward (Static*).

For the No-Control Static routing policy, simulations have been run with different replication limits to show the tendency of the metric's value. We have narrowed the replication limit to the scenario's number of nodes. We consider that having as many copies of the message as nodes are in the network is an approximation of epidemically flooding the network.

Finally, we provide all the obtained results, the ONE configuration files for all the scenarios, the script files to run the simulations, and the data traces for the reproducibility of these results (https://deic.uab.cat/~mcdetoro/controller-driven_OppNet_results.zip, accessed on 3 December 2022).

**Table 3.** Summary of the specific simulation settings per scenario.

| Scenario Setting | Taxis | Info5 | Campus | Emergency |
|---|---|---|---|---|
| Simulation time | 69 h | 70 h | 34 h | 23 h |
| Simulation area | San Francisco Bay | hotel | $4.5 \times 3.4$ km | 1 km$^2$ |
| Mobility model | Contact Traces | Contact Traces | Map-Based + POI | RWP |
| # Nodes | 304 | 41 | 80 | 100 |
| # Contacts | 69,412 | 22,459 | 168,442 | 38,013 |
| TTL (s) | 10,000 | 10,000 | 10,000 | 4000 |
| Buffer size | 10 M | 10 M | 10 M | 10 M |
| Message generation distribution | ISTH | ISTH | ISTH | CBR |
| Message generation frequency (s) | FR: [10–60] s; TR: 1800 s | FR: [10–60] s; TR: 900 s | FR: 10 s; TR: 1800 s | 0–8 h: 10 s; 8 h–end: 80 s |
| Message size | [10–500] k | [10–500] k | [10–500] k | 0–8 h: 500 k; 8 h–end: 10 k |
| Walk speed | N/A | N/A | 0.5 m/s | [0.5–1] m/s |
| Control Settings | Taxis | Info5 | Campus | Emergency |
| Nrof controllers | 10 | 2 | 4 | 20 |
| $[o_{min} - o_{max}]$ | [0.6–0.7]% | [0.3–0.5]% | [0.6–0.7]% | [0.7–0.9]% |
| Additive increase ($k_2$) | 1 | 1 | 1 | 1 |
| Multiplicative decrease ($k_1$) | 0.25 | 0.25 | 0.25 | 0.25 |
| LR nrof inputs ($ž$) | 6 | 10 | 6 | 6 |
| Aggregation interval ($\hat{t}$) | 60 s | 30 s | 300 s | 120 s |
| Prediction time factor ($\phi$) | 2 | 5 | 2 | 2 |
| Directive generation frequency | 900 s | 900 s | 900 s | 900 s |
| Reduction factor for a Decay ($r$) | 0.103 ($d$ of 5% at 1800 s) | 0.3 ($d$ of 1% at 300 s) | 0.058 ($d$ of 5% at 300 s) | 0.3 ($d$ of 1% at 300 s) |
| Decay Threshold | 0.1 | 0.1 | 0.1 | 0.1 |
| nrofAggregations weight ($\alpha$) | 0.2 | 0.2 | 0.2 | 0.2 |

5.5.1. Replication Limit Tendency for the Control Configuration

Figure 6 depicts the tendency of the calculated Control replication limit along the simulation. We observe that the calculated *l* tendency is inverse to the filling up of the buffer for each scenario (Figure 7), i.e., the calculated *l* values are initially high, corresponding to the period where buffers are still not overwhelmed but tend to decrease along the simulation depending on the congestion readings. Specifically, the *l* values for Taxis tends toward 8, Info5 tends toward 2, Campus tends toward 2, and Emergency tends toward 4.

The controller's goal is keeping a high *l*, aiming for a higher delivery ratio and lower latency while the buffers are not stressed, and lowering *l* to prevent this stress from happening. Following this behaviour, for the less congested scenario (Taxis), where the controller can keep a higher *l* value, we observe that the controller decreases *l* slowly. Conversely, for higher congested scenarios (Info5 and Campus), where high *l* values would rapidly overwhelm the buffers, we see that the controller decreases *l* much faster. Specifically, we observe that the controller reduces the *l* in the Info5 scenario faster than in the Campus scenario. Indeed, for the Info5 scenario for all the No-Control configurations, the buffer is similarly overwhelmed, whereas, for the Campus scenario, it depends on the No-Control replication limit configuration. More precisely, we can observe the controller's capacity to adjust the *l* in the Emergency scenario, where in the first half of the simulation (the more congested phase) the controller decreases the *l* and increases it in the second half of the simulation (the less congested phase).

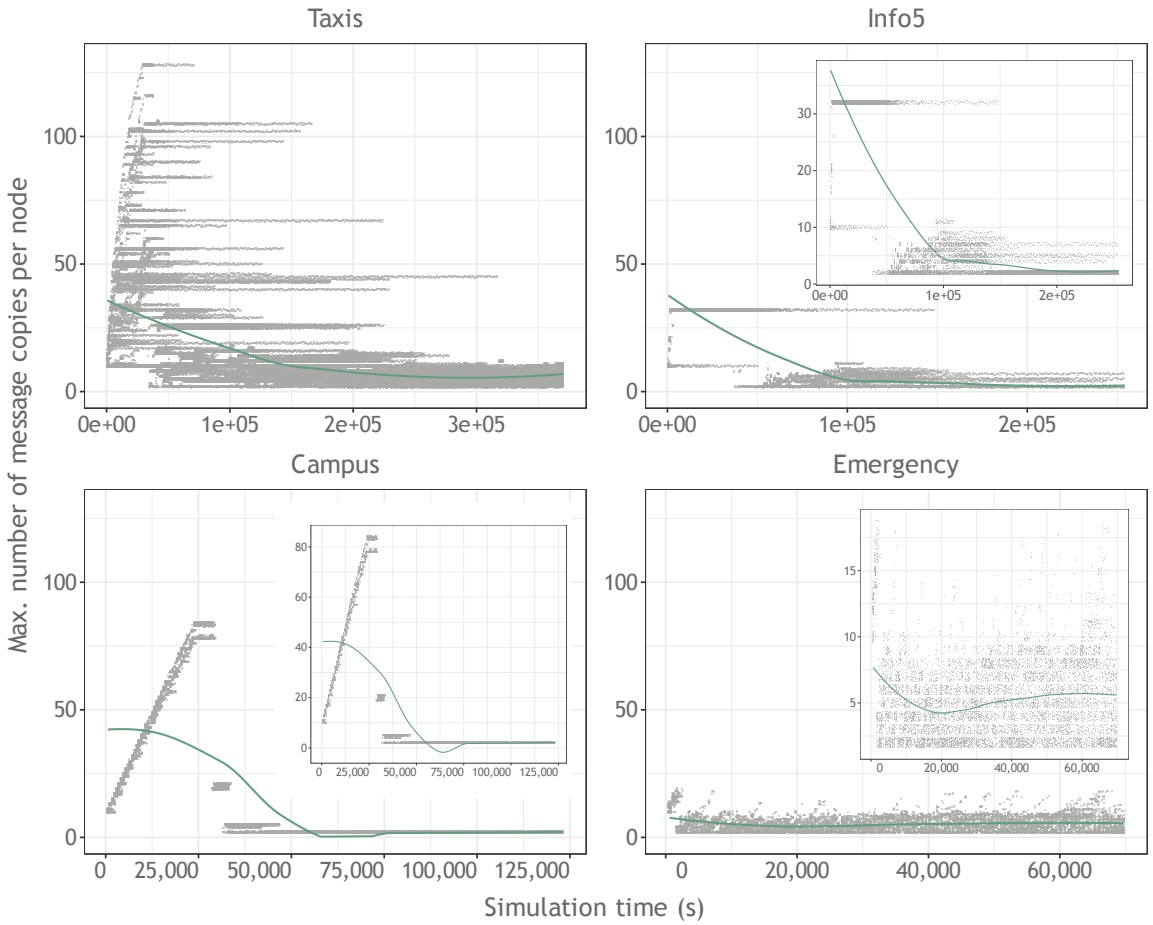

**Figure 6.** For the Control configuration, progression of the replication limit used by each node at each time unit when a new message is created. In blue, we show the tendency of the replication limit over time (for Taxis it is 8, for Info5 it is 2, for Campus it is 2, and for Emergency it is 4). For the scenarios Info5, Campus, and Emergency, a zoomed-in plot with a different scale has been embedded in the main plot.

### 5.5.2. Buffer Occupancy Evaluation

Figure 7 shows that, for all the scenarios and for all the configurations except for the Control configuration, the buffer fills in a logarithmic manner up to the buffer's total capacity. For the Emergency scenario, we can see an inflexion point that derives to a lower buffer occupancy at the simulation time when the message generation distribution changes from high frequency to low frequency.

Without any replication limit ($l$), the EP policy fills up the buffer faster than the other policies. The Static policy's static $l$ determines the speed at which the buffer fills up. For the Control configuration, as the control system regulates the replication limit based on congestion information readings from the nodes, the buffer occupancy fluctuates based on the effects of the new replication limit values.

Overall, the buffer occupancy is lower with the Control configuration. For all the scenarios, after a transient period, the buffer utilization by the Control policy tends toward 21% for the Taxis scenario, 87% for the Info5 scenario, 32% for the Campus scenario, and 22% for the Emergency scenario. Nevertheless, this remarkable difference between the buffer utilization by the Control and the No-Control configurations is due to how we measure the buffer occupancy for the Control configuration. For Control, the buffer occupancy measure does not count the messages that are still buffered but have the flag $\varphi$ set to *false*, so that if buffer space is required, those will be the first messages to be discarded.

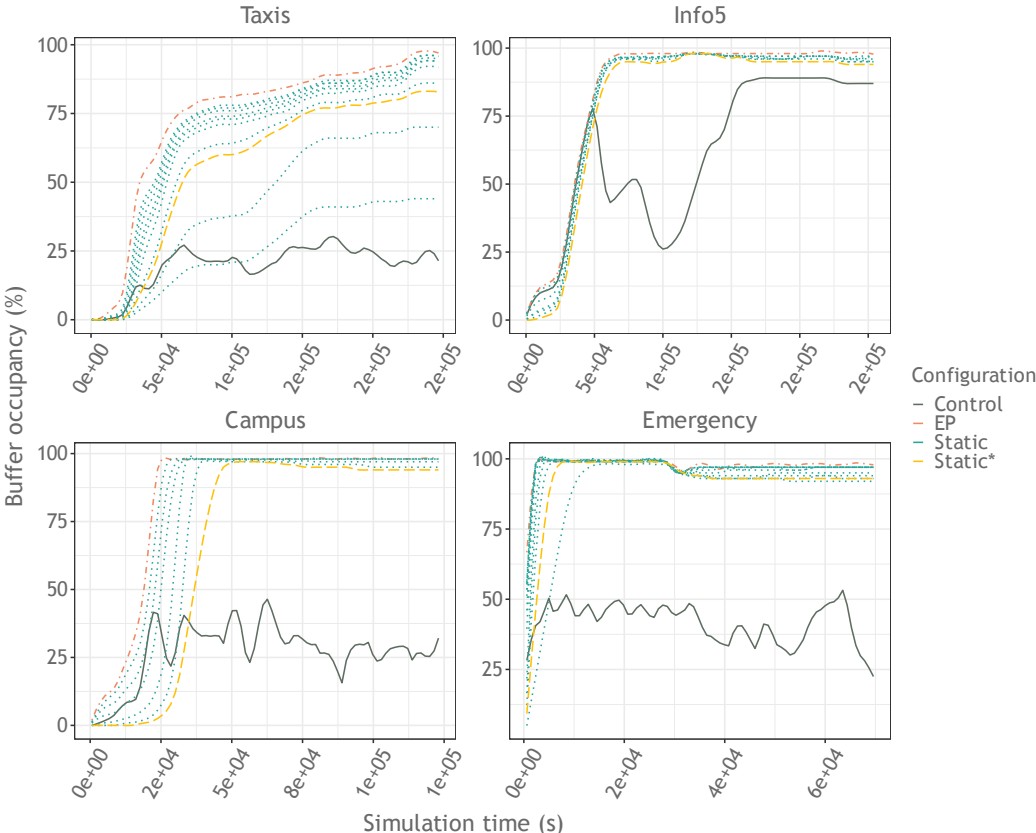

**Figure 7.** Percentage of the buffer occupancy for the different policies, for each scenario, along the simulation time. In the legend, the configuration named Static* corresponds to the Static configuration with the replication limit that the Control tends toward.

Figure 8 shows the percentage of dropped versus relayed messages. As expected, for the No-Control configurations, the faster the buffer fills (Figure 7), the more messages are dropped. Nevertheless, the Control configuration does not have a lower drop rate despite its lower buffer occupation. Precisely, this is because of the aforementioned detail that the Control configuration measures the occupancy of the buffer such that messages that are still buffered but have the flag $\varphi$ to *false* will not be counted when calculating the buffer occupancy, but when the buffer requires the space, they will be dropped. Moreover, as expected, for the more congested scenarios (Info5, Campus, and Emergency), the Control drop rate is slightly higher than that of the Static* configuration. This difference is caused by the changes in the replication limit adjusted by the controller.

More precisely, as expected, for all configurations of the less congested scenario (Taxis), the dropped message rate corresponds to the buffer occupancy, as the buffers are not stressed during the simulation. Furthermore, the relation between the buffer occupancy and the dropped message rate remains for the Info5 scenario, a medium–high congestion scenario, where buffers are more stressed. Nevertheless, for the Campus scenario, a highly congested scenario, and for the Emergency scenario, which has a high-congestion phase, for the Control configuration, the high buffer occupancy ends up dropping the buffered messages with the flag $\varphi$ set to *false*, and, therefore, the dropped message rate is at par with that of the Static* configuration, with slight differences caused by the changes to the replication limit adjusted by the controller.

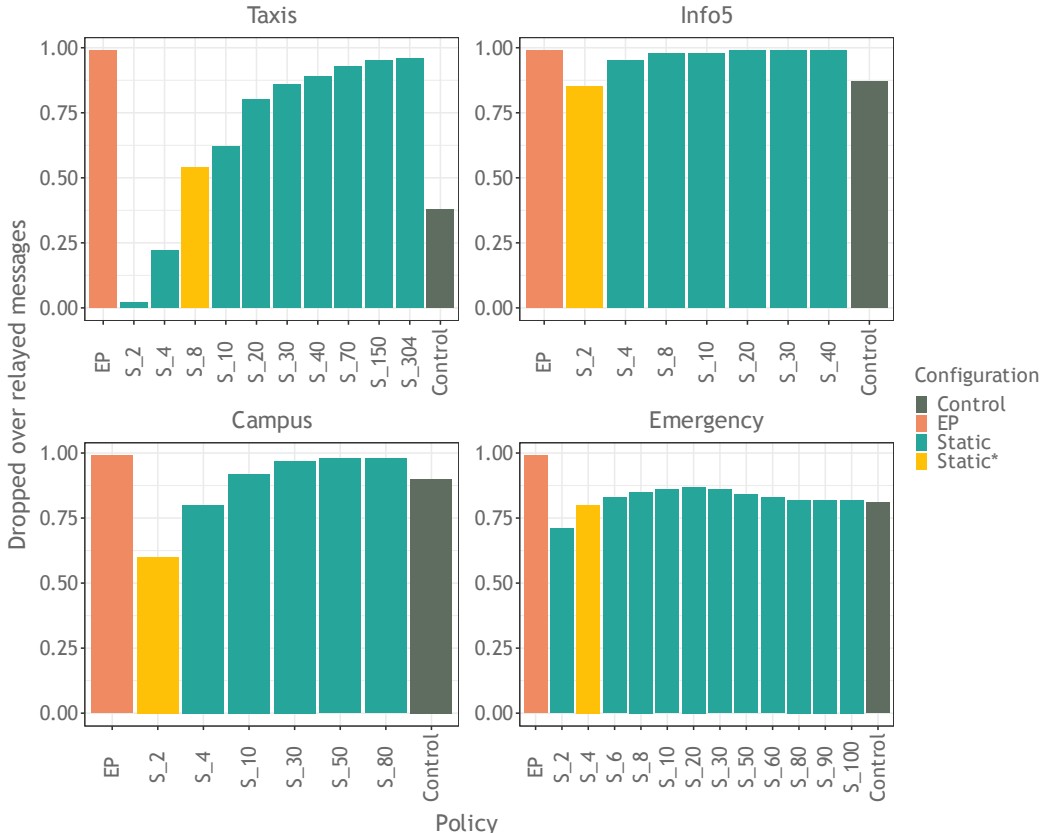

**Figure 8.** Percentage of dropped versus relayed messages by the different policies. The Static suffix denotes the replication limit. In the legend, the configuration named Static* corresponds to the Static configuration with the replication limit that the Control tends toward.

### 5.5.3. Performance Evaluation

This section presents and evaluates the results for the performance metrics listed in Section 5.1 for the Control and No-Control configurations in the different scenarios.

#### Overhead Ratio

Figure 9 shows that the overhead derived from the relay of the message copies depends on the $l$. EP's overhead surpasses Static and Control by two to three orders of magnitude, while Statics's overhead increases for higher $l$ values. The Control's overhead ratio is similar to that of Static*. The slight difference between the two configurations is due to Control's automatic recalculations of $l$.

#### Delivery Ratio

The dropped messages and the overhead directly affect the delivery ratio performance. As we can see in Figure 10, a high replication limit takes its toll on the delivery ratio performance.

With the highest replication limit, the EP policy floods the network, triggering a significant number of drops and, therefore, begets the worst delivery ratio. The behaviour above also applies to the Static policy. The higher the $l$ is, the poorer the delivery ratio we obtain.

Indeed, the Info5 and Campus scenarios (the more congested scenarios), obtain the highest delivery ratio with a Static policy with a low $l$: 2 in both cases. As $l$ increases, the delivery ratio performance decreases. However, for the Emergency scenario, which combines a high message generation frequency with a low one, and for the Taxis scenario, with a low–medium congestion level, the delivery ratio increases for the values of $l$ up to the inflexion point of the Static's $l$ with the best delivery ratio. This behaviour is coherent with

the fact that high values of *l* prejudice the congested scenarios. In contrast, the low–medium congested scenarios admit higher values of *l*, favouring a higher delivery ratio.

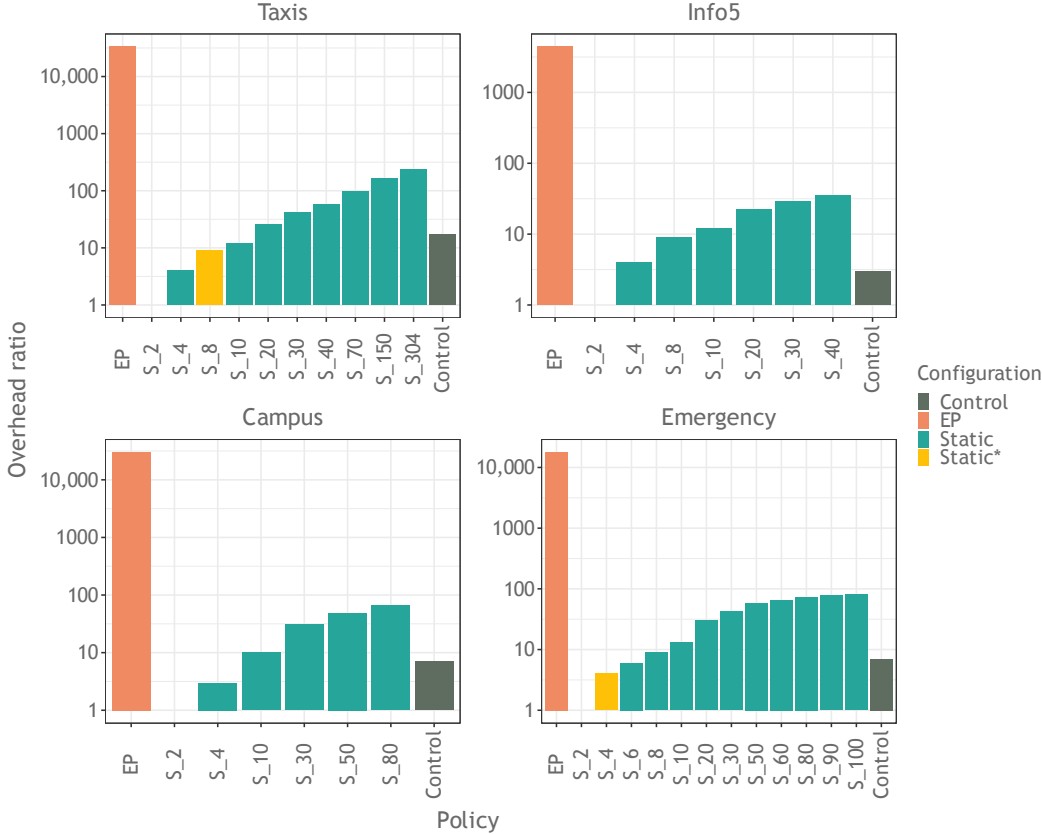

**Figure 9.** The overhead percentage for the different forwarding policies per scenario in a logarithmic scale.

Overall, the Control policy obtains the best delivery ratio for all the scenarios. We obtain the best increase ratio in the scenarios with low–medium congestion levels. We specifically obtain a 14% and an 11% increment in the delivery ratio for Taxis and Emergency, respectively, over the Static configuration with the best-performing *l* value. As we have previously seen, these scenarios admit higher *l* values, bringing on a higher delivery ratio. The ability of the Control policy to dynamically adapt the *l* provides an optimal *l* value depending on the current congestion situation at the current time. This flexibility allows the Control policy to stand out in low–medium congestion scenarios over the other configurations. On the other hand, for highly congested scenarios, where the best option is to keep a very low *l* close to direct delivery, the Control policy provides a low *l* value. It also benefits from the dynamism and slightly outperforms the Static policy with the best-performing *l* value by 4% and 9% for the Info5 and Campus scenarios, respectively.

Latency Average

Figure 11 shows that, despite the crushing effects of the message flooding strategy over the buffer occupancy, delivery ratio, and overhead, when it comes to the latency, message flooding benefits the arrival of the messages to their destination and, therefore, it obtains a good performance. Indeed, as pointed out by Krifa et al. [21], flooding-based replication benefits the latency of the messages at the expense of the delivery ratio in case of congestion. This is because dropped messages will not reach the destination, decreasing the delivery ratio. In contrast, with high message dissemination, the more buffered copies, the more chances that a message copy will have to be delivered upon an opportunistic contact, despite plenty of dropped messages. With this premise, we can see that the configurations that fill up the buffer faster (Figure 7), as the EP and Static configurations with the highest

replication limit perform worst in terms of delivery ratio (Figure 10) but better in terms of latency (Figure 11).

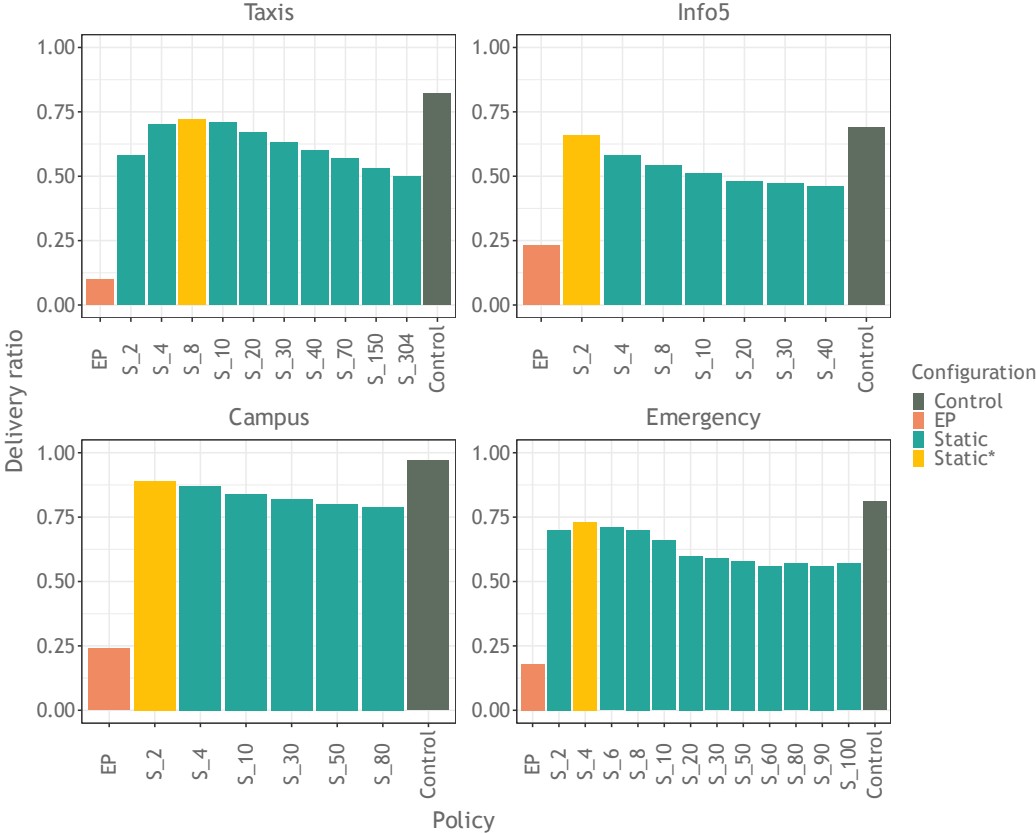

**Figure 10.** Delivery ratio percentage for the different forwarding policies per scenario.

As for the Control configuration, the premise above applies. More specifically, the Control configuration for the less congested scenario (Taxis) achieves a better performance than EP and Static* due to its low buffer occupancy, backed by a low dropped message rate. Nevertheless, the configurations with a static high replication limit leverage from high replication to obtain lower latency than the Control configuration. For the medium- to high-congestion scenario Info5, where the buffer occupancy and dropped message ratio are close to the No-Control configurations, a high replication benefits a lower latency. Furthermore, for the most congested scenarios (Campus and Emergency), where the Control configuration ends up with a high dropped message rate, the highest replication configurations obtain a better latency performance.

Finally, Figure 11 includes the standard deviation of the latency values, revealing that for all scenarios and configurations there is a high variance between the messages' latency values.

### 5.5.4. Evaluation of the Control Settings Impact on the Delivery Ratio

The control layer is configurable through the settings listed in Section 5.4.2. We have run simulations over the four selected representative scenarios to analyze the impact of the Control configuration's settings on the delivery ratio over diverse scenarios and to find a general configuration that fits all of them. The following nine sections present our analysis.

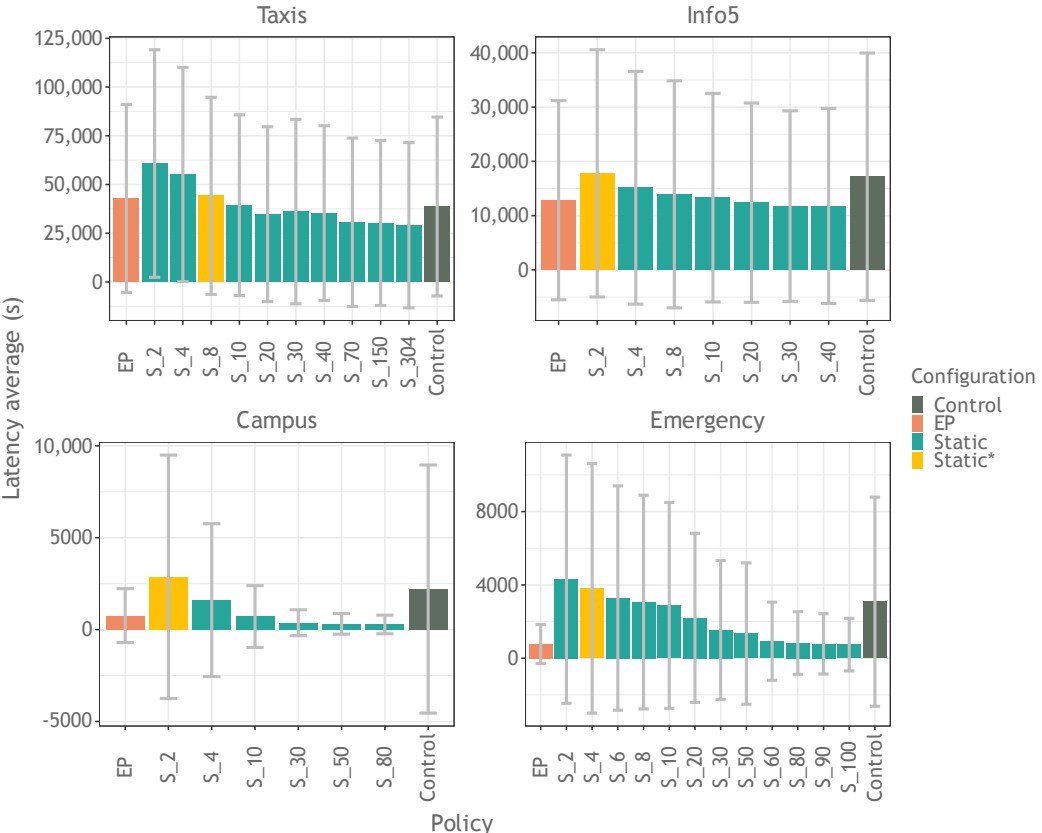

**Figure 11.** Latency average for the different forwarding policies per scenario.

Number of Controllers

Figure 12 shows the delivery ratio depending on the number of controllers used per scenario. In this plot, the max number of considered controllers is 50. Not all the scenarios have been simulated for all the considered number of controllers as, for example, the Info5 scenario has just 41 nodes. Certainly, as pointed out in Section 3.4, the number of controllers needed to orchestrate the OppNet depends on the nature of the network. In addition, the characteristics of the four simulated scenarios, including the number of nodes, are very different. Therefore, for each scenario, we terminated the simulation as soon as the results showed a clear descending slope corresponding to an increasing number of controllers. Specifically, Figure 12 shows that for all the scenarios, a small number of controllers perform better in terms of delivery ratio. For the most connected scenarios, Campus and Info5, the best performance is achieved with just four and two controllers, respectively.

As the controller receives the congestion readings from the nearby nodes, it obtains an overview of the congestion of a part of the network, its nearest part. We can elaborate on this idea by considering that the network is "segmented" by the number of controllers used. Each network "segment" consists of the number of nodes that can be reached by a controller directly or through short time relays.

Having said that, in a highly connected network, using a high number of controllers results in an overlap of the different network segments, as each controller can reach several of these segments. This overlapping effect results in the nodes receiving directives from different controllers. Of course, a directive emitted from a controller from a segment the node does not belong to has congestion information that is not entirely accurate for the node. This overlapping effect is why using many controllers decreases the network performance.

On the other hand, for the sparser scenario (Taxis) and the scenario with an abrupt change in the communication conditions (Emergency), the implicit segmentation derived by the different controllers obtains disjointed segments. Under these circumstances, having

a higher number of controllers (10 for Taxis and 20 for Emergency) helps cover a broader network range, translating to better performance.

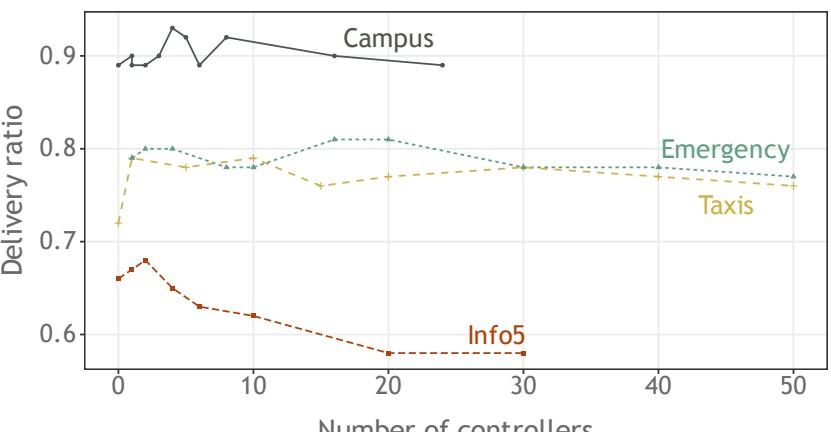

**Figure 12.** Delivery ratio by number of controllers.

Optimal Congestion Interval ($o_{min} - o_{max}$)

As presented in Section 4.4, the current replication limit is not modified when the congestion calculated by a controller falls in the optimal congestion interval.

Figure 13 shows that for the most congested scenario (Campus), there is a similar delivery ratio for all the optimal congestion intervals. The variance between the performance results for the different intervals is just 0.0002. Nevertheless, we obtain the best result for the interval 0.5–0.8.

These homogeneous results infer that in a congested scenario, if the congestion predictions fit in the configured optimal congestion interval setting for the current replication limit, the best strategy is to keep the current replication limit steady.

The medium- to high-congestion scenario (Info5) shows slightly more variance in the delivery ratio than the previous scenario (0.001). This variance can be appreciated mainly when the interval upper bound ($o_{max}$) is 0.8 and 0.9. Therefore, when the optimal congestion interval upper bound is set close to the maximum buffer occupation, the replication limit set by the controller is too high. Thus, a more conservative optimal congestion range gives better results which, in this case, is 0.6–0.7.

For the Taxis scenario, the sparsest scenario with fewer contact opportunities, it can be seen that the most conservative interval configuration, 0.3–0.5, performs by far the worst (24% less than the best interval). In contrast, the intervals with a high $o_{max}$ have a good performance. This is because using a high level of message replication promotes a higher message delivery in a sparse scenario.

Finally, the Emergency scenario behaves similarly to the Taxis scenario. The performance of the most conservative interval is the worst (11% less than the best range). Nevertheless, the performance variance of the different intervals is 0.0004, whereas for the Taxis case it is a bit higher: 0.002. For an unpredictable scenario, similarly to the Taxis scenario, the best strategy is to use an interval with a high $o_{max}$ to keep a high replication limit and, therefore, to have more chances for message delivery.

Altogether, we have seen that in a congested scenario, the key is to maintain a steady replication limit if it maintains the congestion within the configured optimal congestion interval. It is better to set a conservative optimal replication limit for a medium- to high-congestion scenario to avoid high replication that could yield in a future congested scenario. On the contrary, for low-congestion scenarios, setting an optimal congestion interval with a high upper bound leads to a higher replication limit favouring message replication, which increases the delivery ratio.

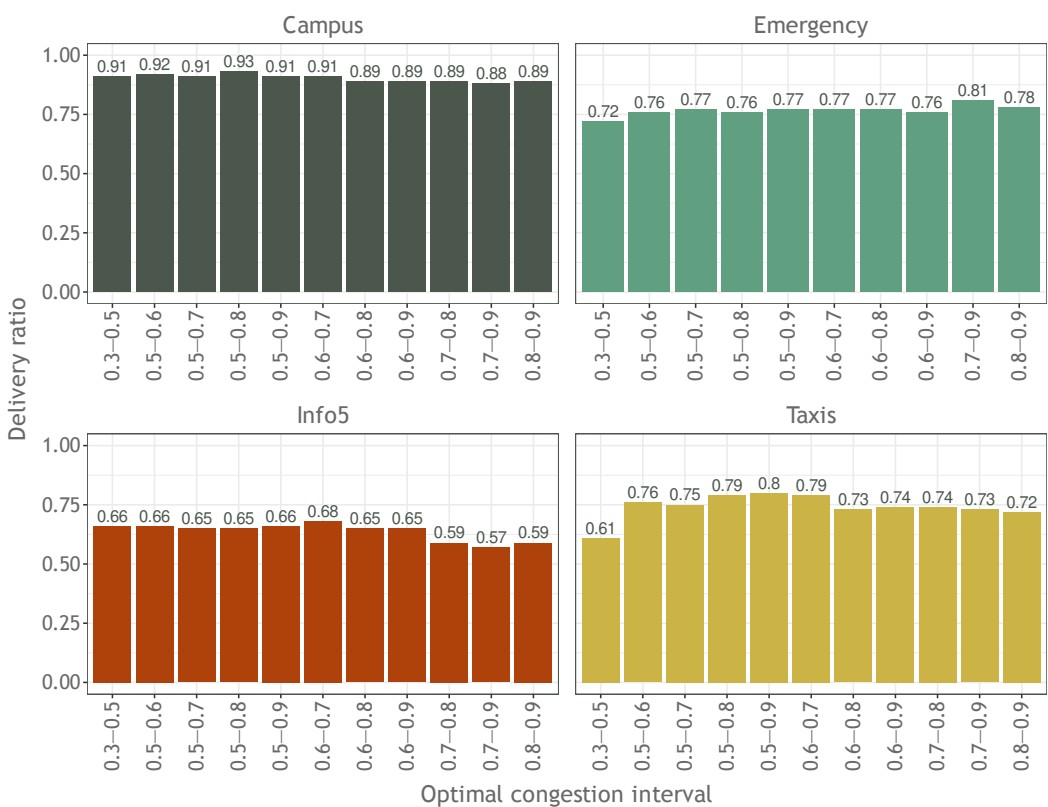

**Figure 13.** Delivery ratio depending on the optimal congestion interval per scenario.

Additive Increase ($k_2$); Multiplicative Decrease ($k_1$)

Figure 14 shows that for the additive increase factor used in (7), the value that gives the best performance in all the scenarios is, undoubtedly, 1. From this result, it can be stated that it is essential that the replication limit grows slowly to mitigate the adverse effects of high replication as much as possible. As for the multiplicative decrease factor (MD), for all the scenarios except for the Campus scenario, the best option is to reduce 75% the replication limit as a drastic measure to decrease the congestion caused by replication.

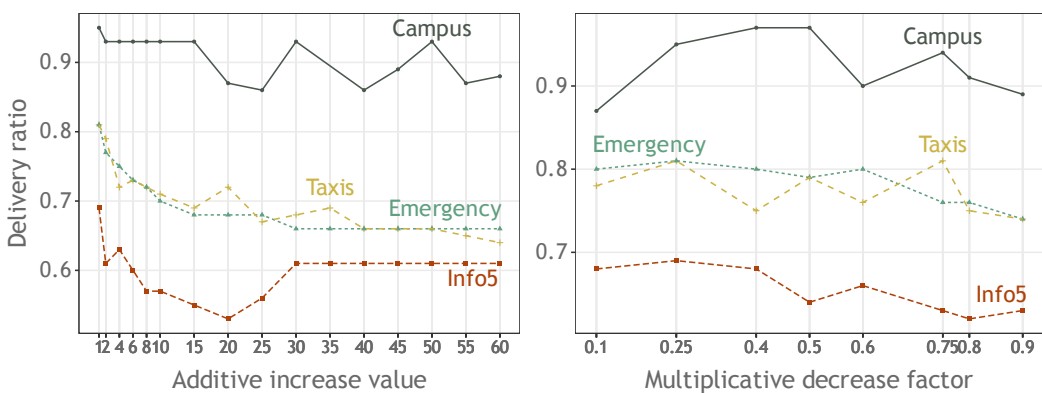

**Figure 14.** Delivery ratio vs. different values for the AIMD control function.

For all the scenarios except for the Taxis scenario, the delivery ratio continues decreasing for higher MD values (implying less replication reduction). For the particular case of the Taxis scenario, its sparse condition results in more fluctuating delivery ratios depending on the MD factor, but none overcome the ceiling achieved with the 75% of reduction.

Going back to analyzing the results for the Campus scenario, which is highly connected and the most congested, we can see that we obtain the best result by applying a reduction

factor of 50%, maintaining a higher replication level. These results are consistent with those in the previous section, where it was stated that once the congestion status fitted in the congestion range thresholds, the best strategy was to keep the replication limit within the range. Precisely, reducing the replication limit to half is a good method for keeping the congestion steady within the optimal congestion interval.

Aggregation Interval ($\hat{t}$); Number of Inputs ($\check{z}$)

Figure 15 shows that gathering congestion readings during a short time interval ($\hat{t}$) results in obtaining the best performance for all the scenarios. Hence, we can say that the best results are obtained when the controller generates directives at a higher rate. There are minor differences between the suitable intervals for each scenario.

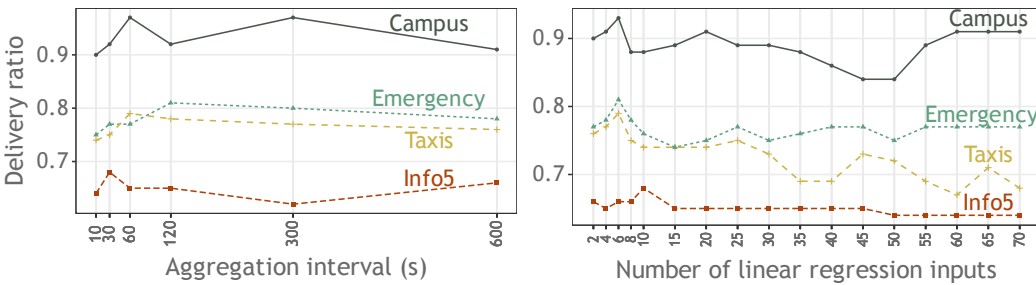

**Figure 15.** Left: delivery ratio vs. the time interval while the controller is gathering congestion readings. Right: delivery ratio depending on the number of inputs the linear regression is fed with.

The best performance for the Taxis and the Campus scenarios is at a 60 s interval, and for the Info5 scenario (the scenario with the fewest nodes) is 30 s. Thus, it can be said that the fewer nodes there are, the higher rate of directives emitted by the controller is required. For the Info5 scenario, even though in the plot it seems that from the interval of 600 s the function is increasing, simulations up to an interval of 10,800 s with samples every 1800 s have been run, and the delivery ratio remains constant to the value obtained at the 600 s interval.

Finally, the best interval time for the Emergency scenario is 120 s. This scenario drastically changes the message generation in the middle of the simulation. In that variable situation, it is understandable to have a wider time interval range for gathering congestion readings to mitigate the effects of the changes.

On the other hand, regarding the number of entries in $\check{M}$ ($\check{z}$) used to calculate the congestion prediction, Figure 15 clearly shows that, for all the scenarios it is better to have a small $\check{z}$. This is fully understandable, as the fewer inputs we use, the newer the information is.

Prediction Time Factor ($\phi$)

As shown in Figure 16, a minor factor, i.e., predicting the future congestion in the short term, gives the best performance for all the scenarios. For all of the scenarios, the factor is 2 except for that of Info5, which is 5. Hence, we can determine that we can predict the near future more accurately than the far future. This assertion implies that we need a small factor combined with a small aggregation interval ($\hat{t}$). This combination is feasible as, in the previous section, we have seen that a small $\hat{t}$ leads to better performance.

Reduction Factor for a Decay ($r$)

In Figure 17, we consider different decay at different times (300 s, 600 s, 1800 s, and 3600 s). The decay ranges from 1 (no decay) to (0.01) at each considered time. For all the scenarios, we can see that the delivery ratio overwhelmingly drops when the congestion reading is not "penalized" by a decay (decay weight close to 1) after an elapsed time. Hence, we conclude that considering a decay for the congestion readings is crucial.

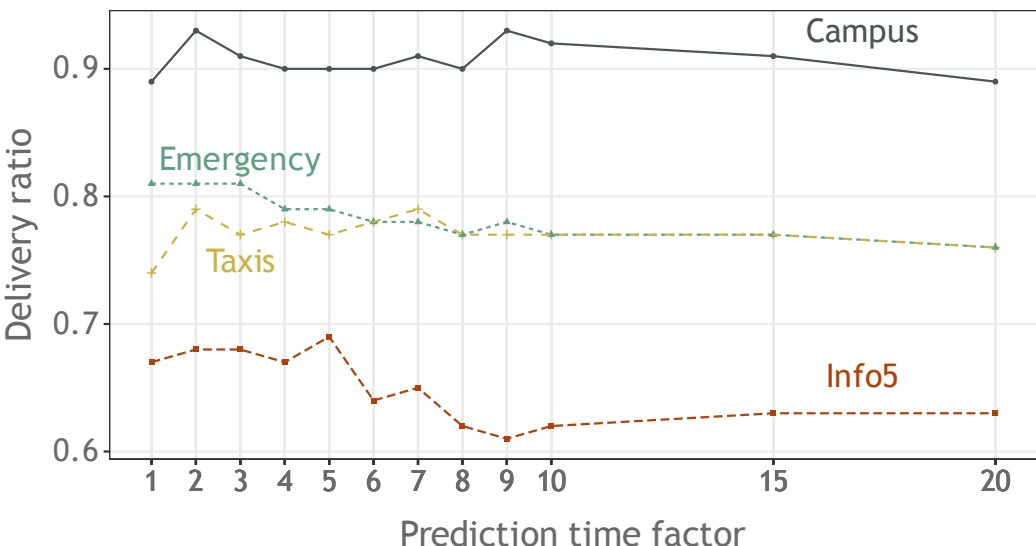

**Figure 16.** Delivery ratio depending on the prediction time factor.

From the transient of the results, for each scenario, it can be seen that the decay at *t* heavily affects the performance. Nevertheless, it can be observed that for high decay "penalties" (small decay weight values), better results are obtained at any time than with lower decay (high values).

In conclusion, the best strategy is to apply a high decay (small weight factor value), which implies a considerable reduction in the effect of the congestion measurement after a short elapsed time from its creation up to its reception by a node/controller.

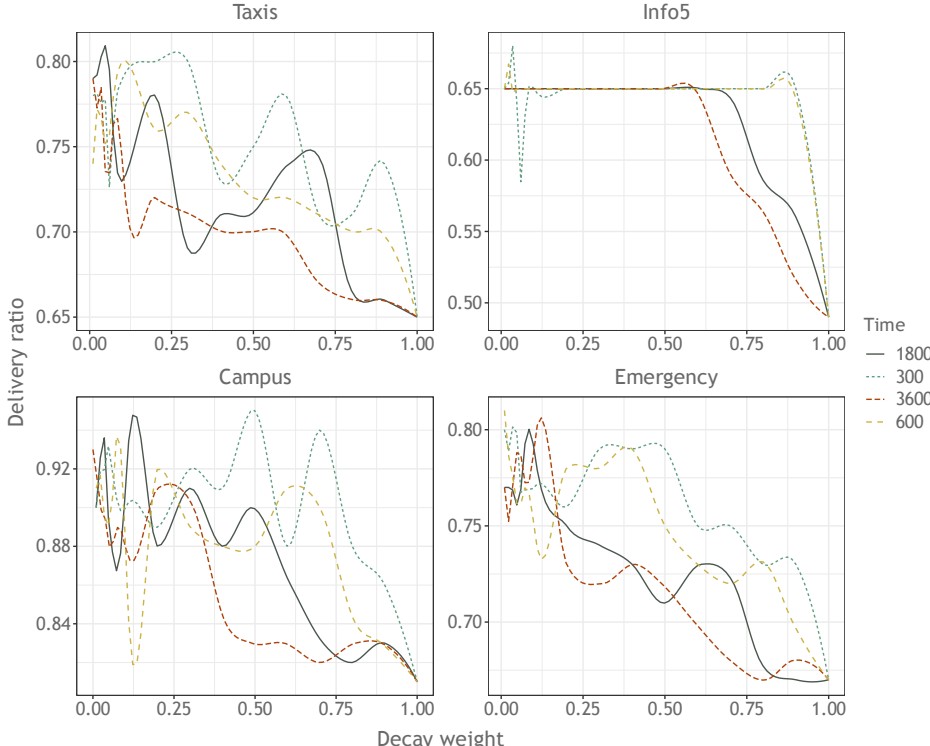

**Figure 17.** Tendency of the delivery ratio when combining different decay percentages at different times per scenario.

Decay Threshold

Section 3.3 shows how the received measurements are aggregated through (3). This equation double weights the congestion reading by the number of aggregations it is formed by and by its decay, which is calculated with (1). Consequently, a congestion measurement formed by a high number of aggregated congestion measurements would have a high impact, despite its decay in the overall process of the congestion measurements aggregation. Hence, several of these congestion readings in the control's current aggregation process can lead to a long tail effect [39], where almost negligible old congestion readings would highly affect the whole aggregation result. In this case, we would have a congestion reading calculation based on, very likely, expired information. To avoid this undesirable situation, a decay threshold is specified so that the congestion readings with a decay smaller than decay threshold are not considered in the controller's congestion calculation. From Figure 18 it can be stated that, for all the scenarios, the best decay threshold is 10% of the decay.

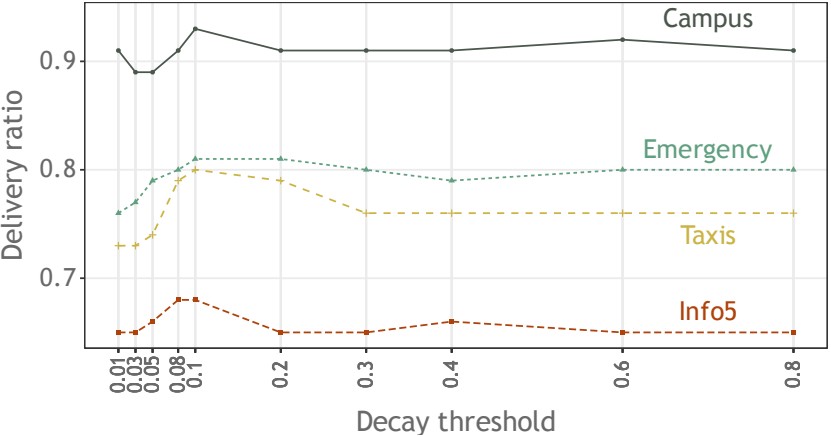

**Figure 18.** Delivery ratio for different *decay thresholds*.

Figure 18 shows that, as expected, either aggregating old readings (small decay threshold) or discarding new readings (high decay threshold) worsens the performance consistently for all the scenarios. Nevertheless, although small decay thresholds diminish the performance for the Emergency scenario, high decay thresholds do not significantly affect such performance. This behaviour is due to the high variance in the latency of this scenario, so considering old readings does not significantly affect the performance.

Number of Aggregations Weight ($\alpha$)

The number of aggregations weight setting ($\alpha$) used in (3) is the weight applied to the *number of aggregations* an aggregated congestion measurement is formed of. As (3) is a two-factor weighted average, the weight related to the *decay* is the *alpha*'s complementary $(1 - \alpha)$.

As we can see in Figure 19, for all the scenarios, we obtain the best performance with an alpha of 0.2. This result concludes that the decay of a congestion measurement is more relevant than the number of aggregations this aggregated measurement is formed of.

Directive Generation Frequency

As we can see in Figure 20, for all the scenarios except for the Taxis scenario, we obtain the best performance by resending the last directive each 900 s, provided no contact has happened before. Nevertheless, for the Taxis scenario, we obtain the optimal performance with a directive frequency of 300 s. We can understand this slight difference, as the Taxis scenario is the sparsest scenario, which implies fewer contacts between nodes. Hence, a more frequent directive beckoning gives the nodes more chances to receive a directive, and consequently, we get better performance.

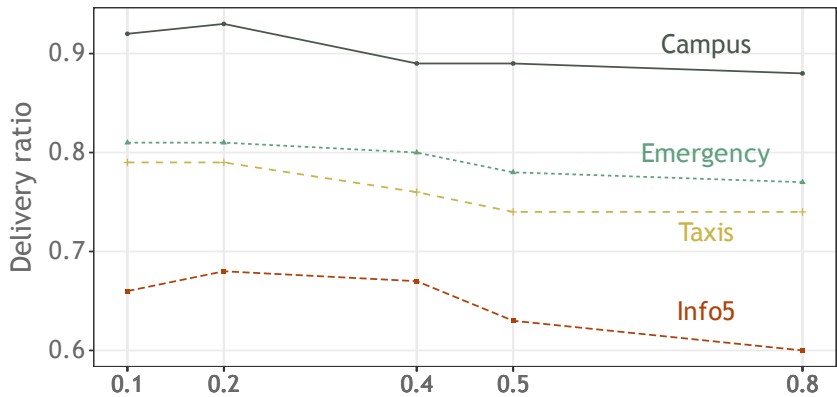

**Figure 19.** Delivery ratio for different $\alpha$ weights.

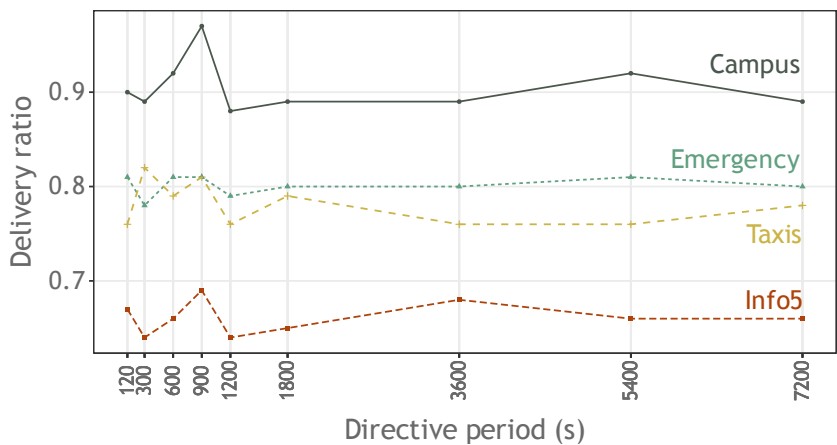

**Figure 20.** Delivery ratio for different directive generation periods.

## 6. Conclusions

The motivation for this work was bringing the benefits of the SDN architecture into OppNets by having a controller to retrieve network context information to orchestrate the data plane in terms of tuning the forwarding strategy to achieve a better network performance.

To state the soundness of the proposal, the control layer has been applied to manage the congestion derived from multi-copy-based forwarding algorithms. This controller-driven OppNet has been tested over four scenarios characterized by different mobility patterns and node densities against baseline forwarding strategies based on message replication.

For the scenarios with a message distribution following the ISTH function, the simulations show that for the Control configuration, the replication limit tends toward an asymptote proximal to the replication limit of the best-performing Static policy in terms of delivery ratio (optimal). Therefore, it can be stated that under blind network knowledge, the Control configuration approaches the *optimal* replication limit.

Moreover, the Control configuration adapts to changes in the pattern of message generation distribution. It is precisely under these unpredictable conditions that the Control configuration stands out over the other configurations by leveraging its dynamic adaptability to the network conditions. This adaptability facilitates a significantly lower occupancy of the node's buffer and an important reduction in the overhead intrinsic to replication.

Furthermore, the Control configuration improves the delivery ratio for all the scenarios. Its effectiveness is accentuated for scenarios with medium–low congestion, as a wider replication limit range can be considered. In contrast, a highly congested scenario is stuck

to a low replication limit. Undoubtedly, latency benefits from a replication that does not overwhelm the nodes' cache system. The fact that the Control configuration keeps the replication limit at bay to avoid congestion and achieve a better delivery ratio affects the latency. Therefore, the application layer should determine whether to maximize the delivery ratio or minimize the latency, so the Control configuration could apply forwarding strategies to optimize one or the other.

Furthermore, the control layer is highly configurable to provide the best performance depending on the Oppnet's nature. Nevertheless, generic values providing a good performance have been determined from simulations over the aforementioned scenarios. In this regard, simulations show that the controller must use recent congestion readings from contacts. Therefore, applying a decay weight over the measurements used to predict the network conditions is decisive. In this regard, it is more effective to perform a short-term prediction than a long-term prediction.

Moreover, the simulation demonstrates that the sparser the network is (fewer contacts between nodes), the more directives are needed. For the use case of congestion control, the replication limit needs to grow slowly, whereas, in a congestion state, a sharp reduction is required. The optimal congestion interval is highly coupled to the characteristics of the scenario.

Finally, simulations depict that, despite the disconnections, network partitioning, and long delay paths prone to OppNets, a small number of controllers suffices.

Overall, this study asserts that (i) a context-aware system built upon the SDN pillar principles is a good approach for context-management in OppNets and (ii) using this context-aware system to regulate the replication in an OppNet driven by a multi-copy forwarding strategy leads to better network performance.

**Author Contributions:** Conceptualization, C.B. and S.R.; data curation, M.d.T.; formal analysis, M.d.T.; funding acquisition, S.R.; investigation, M.d.T.; methodology, M.d.T.; project administration, C.B.; resources, C.B.; software, M.d.T.; supervision, C.B.; validation, M.d.T.; visualization, M.d.T.; writing—original draft, M.d.T.; writing—review and editing, C.B. and S.R. All authors have read and agreed to the published version of the manuscript.

**Funding:** This research was funded by The Catalan AGAUR 2017SGR-463 project and The Spanish Ministry of Science and Innovation PID2021-125962OB-C33 project.

**Institutional Review Board Statement:** Not applicable.

**Informed Consent Statement:** Not applicable.

**Data Availability Statement:** The control layer extension for the ONE simulator is available at https://github.com/MCarmen/the-one/tree/control, (accessed on 3 January 2022). The obtained results, the ONE configuration files for all the scenarios, the script files to run the simulations, and the data traces for the reproducibility of these results are available at https://deic.uab.cat/~mcdetoro/controller-driven_OppNet_results.zip, (accessed on 3 January 2022). The scenarios based on real mobility traces, Taxis and Info5, are also available at https://crawdad.org/epfl/mobility/20090224/ and http://crawdad.org/uoi/haggle/20160828/one, (accessed on 3 January 2022), respectively.

**Acknowledgments:** We want to thank the guidance and coaching of Ian Blanes Garcia with the writing of this manuscript, Cristina Fernandez Córdoba for showing us the beauty of mathematical formulation, and Joan Borrell Viader for accompanying us in the early stages of this research.

**Conflicts of Interest:** The authors declare no conflict of interest.

## Abbreviations

| | |
|---|---|
| $\mathfrak{a}$ | Application layer message |
| $\varphi$ | Message's flag alive |
| $b$ | Buffer size in bytes |
| $B$ | Buffer: a list to store messages |
| $\mathfrak{c}$ | Controlled variable |
| $c_{t+n}$ | Congestion state prediction for time $t+n$ |
| $d$ | Decay |
| $\delta$ | Directive: $\delta = (\mathit{Id}, \vartheta)$ |
| $\delta_l$ | Directive encapsulating a replication limit |
| $\phi$ | Factor to calculate $t_{t+n}$ |
| $\mathit{Id}_p$ | Identifier of the network setting $P$ |
| $\mathit{Id}_l$ | Message's replication limit Id setting |
| $k_1$ | Multiplicative decrease factor |
| $k_2$ | Additive increase factor |
| $g_i$ | A message |
| $\#g_d$ | Number of delivered messages |
| $\#g_c$ | Number of created messages |
| $\#g_r$ | Number of relayed messages |
| $l$ | Max. message copies in the network (replication limit) |
| $\overline{\lambda}$ | Latency average |
| $\lambda_i$ | Latency of message $g_i$ |
| $L_1$ | Highest message generation rate for the ISTH dist. |
| $L_2$ | Lowest message generation rate for the ISTH dist. |
| $m_{l_i}$ | Node $n_i$'s local network measurement: $m_{l_i} = (v_{l_i}, 1, t_{c_i})$ |
| $m_i$ | Aggregated received network measurements: $m_i = (v_i, \eta_i, t_{c_i})$ |
| $m_{t+n}$ | Predicted network measure value at time $t+n$ |
| $M_i$ | Aggregated network measurements list for node $n_i$ |
| $\breve{m}$ | Aggregation of the received measurements during $\hat{t}$s |
| $\breve{M}$ | List of $\breve{m}$ values |
| $\breve{z}$ | Max size of $\breve{M}$ |
| $\mu$ | Control system's manipulated variable |
| $\eta$ | Number of aggregated measurements |
| $n_i$ | A specific node |
| $o_i$ | Buffer occupancy in bytes of node $n_i$ |
| $o_{t+n}$ | Buffer occupancy prediction for time $t+n$ |
| $o_{min}$ | Min. buffer occupancy rate |
| $o_{max}$ | Max. buffer occupancy rate |
| $\theta$ | Overhead ratio |
| $r$ | Reduction factor |
| $\mathfrak{r}$ | Control system's reference input |
| $\varrho$ | Number of times a message has been relayed |
| $\rho$ | Linear regression function |
| $\sigma$ | Delivery ratio |
| $t$ | Current time |
| $t_{t+n}$ | Time ahead when calculating the congestion prediction |
| $t_{c_i}$ | Creation time for either a measurement or an aggregation |
| $\hat{t}$ | Time period for aggregating received measurements |
| $\breve{T}$ | List of the timestamps of the aggregations performed |
| $\vartheta$ | Network setting's value |
| $v_{l_i}$ | Node $n_i$'s local network measurement |
| $v_i$ | Result value after aggregating the received measurements in $M_i$ |

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
