# Peer review of "A Controller-Driven Approach for Opportunistic Networking"

_applsci, doi:10.3390/app122312479_

Round 1

Reviewer 1 Report

This paper is well-written and well-organized.

The work presented addresses a current scientific interest topic.

The developed algorithms are very detailed and clearly presented.

References are updated and complete.

In figure 6, the vertical scale of the graphs should be the same for all to facilitate the understanding of the data presented.

The graphs in figure 6 are all very different. There could be a brief explanation of the shape of the graphs to facilitate your interpretation.

Figure 7 shows the percentage of buffers occupancy for all configurations. There is a high buffers occupancy which greatly impacts latency and can also reduce throughput.

We can say that all configurations have poor performance, with the exception of the Control configuration, because they allow buffer occupancy to always be close to 100%. It is possible that the parameterization of the tests was not optimal. Control configuration keeps buffer occupancy much lower than in other configurations. This is a positive sign.

Figure 8 shows the average number of messages discarded. These data are not clear because we do not know the total number of generated messages. This graph should be presented as a percentage.

Comparing the graphs in Figures 7 and 8, it would be expected that the messages discards would be much higher in configurations with high buffer occupancy rates. This is not verified.

In figure 10 the graphs should be presented with the same scale for the various configurations, to facilitate the interpretation of the results. This data shows the high performance of the Control configuration.

The data presented in Figure 11 (latency) are contradictory to the data presented in Figure 7 (buffer occupancy). For example, the EP configuration shows buffer occupancy close to 100%. This means that messages spend a lot of time in buffers and this will have a negative impact on latency. This is not confirmed by Figure 11 data.

The Control configuration should have lower latency than the other configurations because in the graph of Figure 7 the buffer occupancy is lower than the other configurations.

Part of the text of the conclusion is a summary, not a conclusion. So the Conclusion section could be simplified.

Reviewer 2 Report

The authors propose a mechanism for efficient message dissemination / routing in Opportunistic Networks, in which nodes use opportunistic contacts with other nodes to forward messages, based on a so-called GAFA strategy (for "gathering, aggregating, and forwarding").

Comment: The goal of the proposed mechanisms and protocols is not clearly stated in the paper (although it can be inferred from the different parts of their proposal), and should be better identified. Their first contribution, stated in section 1 (“novel device context-aware system for OppNets inspired by the SDN architecture”, lines 62-63), does not provide sufficient insight about the purpose of that system, or the conditions in which the network operates (beyond being an OppNet). Data forwarding (line 79) and context-based routing are mentioned (line 88), but it’s not clear if general routing (i.e. providing routes between all possible pairs of nodes in the network) is the final purpose of their proposal.

They consider two types of nodes in the network (inspired in the SDN architecture): controllers and agents. There can be several controllers in a network, each having a partial view of the network (sec. 3.4). Agents collect measurements, share them to other nodes upon contact, and aggregate them. Controllers receive measurements and aggregate them (as agents), and also generate “directives” (sec. 3.1.2) that modify network settings (locally configurable by nodes), and that are disseminated through the network in the same way. 

Comments:

It is suggested to clarify their terminology and ensure that it used consistently throughout the paper. The term “node” sometimes means a generic entity in the network, and sometimes a non-controller entity of the network (see e.g. Fig. 1); in other cases, the latter is denominated “SDN-like agent”.

There is a confusion on line 199 (or, it is unclear): “Any node could potentially be a controller. Which nodes act as controllers depend on the nature of the network. In a vehicular network (VANET), the nodes [controllers?] could be the roadside units; in an information-centric network, they could be well-connected nodes.”

It is not clear which directives are used and how they modify the network performance. From lines 309 and next, generic “directives” are modified according to an AIMD strategy, which suggests (although it’s not stated at this point) that the considered directive describes the queue length. Lqter, in section 4, this generic case is particularized for “congestion control”, and directives consist of node queue/buffer length.

Directives are generated opportunistically (as stated initially, denominated “asynchronously”) and periodically (denominated “synchronously”). “Asynchronously generated” directives are supposed to be controller reactions to collected context measurements. The paper considers (section 3.5) a prediction mechanism consisting on linear regression, to generate periodic directives.

Comment: Terminology is misleading here. “Asynchronous” corresponds to “non-periodically” or “opportunistically”, since these generation depends on opportunistic contacts (it could be actually said that it’s “synchronized” to the contact between nodes). What is called “synchronous” is really “periodic” (“i.e., every time period”, as stated in line 363), but there is no need of network time synchronization (=time agreement) for that (only autonomous detection of time periods at each node). 

The proposal is specified in general in section 3, and adapted for the “congestion control” functionality in section 4. This case is evaluated through simulations in section 5, over synthetic traffic models (sec. 5.3) and four mobility scenarios (sec. 5.4), some synthetic, some from the well-known CRAWDAD database. Evaluation metrics (sec. 5.1) are standard (delivery ratio, overhead, etc.). Proposed approach is compared with Epidemic Routing from Vahdat et al. (2000), and Static quota-based replication algorithm, from Spyropoulos et al. (2005). 

Comment: The basic operation of the approaches to compare should be minimally presented in sec. 5. Since both are relatively old proposals (presented in 2000 and 2005), the election of these proposals for comparison should be justified in terms of relevance, or interest of the relative evaluation.

Presented results show that the proposed scheme gets better buffer occupancy (Fig. 7), with similar or slightly better performance than static configurations, and substantially better performance than the epidemic mechanism. Observations highlighted in the conclusions are intuitive and reasonable.

Comments:

In Fig. 6 (sec. 5.5.1) it can be distinguished two regions, probably corresponding to the warm-up period, specific of each network. It would be worth to explain these different warm-up behavior.

In Fig. 8, and in general in averaged results, it would be helpful to indicate variance measures (or 95% confidence) for each value.

Scientific points

The paper proposes a fully decentralized system that allows messages to be efficiently disseminated through an Opportunistic Network, and the network to be self-configured through the dissemination of “directives”. From the main results presented in section 5, it can be observed that the performance of the proposed approach is not very different in average to the performance observed for the static mechanism with a replication limit corresponding to the one dynamically identified by the proposed mechanism. In that sense, the proposal seems to be oriented to the dynamic agreement on a shared value (e.g., a replication limit for congestion control and optimized dissemination).

The proposal is inspired by the SDN architecture. Since several controllers are allowed in the network, and they may produce inconsistent directives, it would be interesting to elaborate on how these inconsistencies should be handled in the network, which policies could be used, and what would be their implications in different scenarios, depending on the characteristics of the OppNet.

The paper proposal includes a large amount of choices and configuration parameters (decays, aggregation weighting, AIMD parameters, congestion intervals, prediction models and parameters…). Some of these choices (e.g. the use of linear regression for prediction, the choice of an AIMD scheme for updating values) Evaluation through simulations introduces a large set of additional parameters (traffic pattern, mobility synthetic patterns…). In the presence of that many parameters, an in-depth discussion should be provided so as to understand (i) the (qualitative) impact of system parameters in the performance, and (ii) the relevance of the presented (synthetic) simulations, i.e., whether obtained results are expected to change substantially, should some of the key simulation parameters be very different. 

General presentation points

The paper would benefit from a large editorial iteration, so that scientific relevance of the contributions can be properly assessed. While the English is correct, the document is unnecessarily verbose (with several redundant observations) and unclear in several key points. The particular problem that is addressed is not clearly specified and is hard to identify, the approach (described in sections 3 for the general case, and 4 for the particular congestion control case) is difficult to infer. 

Partly related to this, the structure could be improved. The generic approach presented in section 3, for instance, in which directives are not precised, does not allow to characterize directives features or dynamics of the “directives”; the choice of a P-controller with AIMD to adjust the value seems then unmotivated. This is a natural choice in section 4, in which the directive affects the replication limit of the packet (line 412): the pertinence of the AIMD strategy is associated to the type of directive (in this case, replication limit) under consideration, and should be then introduced at this point.

Similarly, it is suggested, in order to improve readibility of the paper, (i) to present together the main principles of the proposed approach, before developing each of them in detail, (ii) to ensure terminology consistency, avoid redundancies in the text, and shorten the paper, which is very verbose. 

Reviewer 3 Report

This paper is excellently written, and all of the network scenarios that were conceptualized are described in a clear and concise manner throughout the paper.

1. What is the main question addressed by the research?

Considering that congestion problem is very critical in any networked system, using "a controller-driven design," the authors of this paper try to alleviate a pressing issue in modern networking system

 2. Do you consider the topic original or relevant in the field? Does it address a specific gap in the field?

The issues addressed in the paper is interesting, and it does touch on the data flow in a decentralised network, where this is important for the functioning of the networked system

3. What does it add to the subject area compared with other published material?

Utilizing the controller's given context information to configure the forwarding protocol increases the delivery ratio while maintaining a good latency average and a low overhead,

4. What specific improvements should the authors consider regarding the methodology? What further controls should be considered?

The simulation reflecting both actual and simulated mobility traces, in attempt to solve the critical congestion problem in networked systems with "a controller-driven architecture" is praiseworthy. Futhermore, the simulated mobility traces that shows that using the controller's given context information to configure the forwarding protocol increases the delivery ratio, keeps the average latency low, and requires little extra work. Doing so contributes to resolving the severe congestion problem that currently characterises today's networks. In the context of efficient communication, that is crucial.

5. Are the conclusions consistent with the evidence and arguments presented and do they address the main question posed?

The simulation, which is based on both real and simulated mobility traces, is consistent with the evidence and arguments presented, and it does address the main question posed in an effort to resolve the critical congestion problem in networked systems: whether or not using the controller's provided context information to configure the forwarding protocol improves the delivery ratio while keeping the average latency and overhead to a minimum.

6. Are the references appropriate?

All of the citations make sense and are appropriate

7. Please include any additional comments on the tables and figures.

The entirety of the numbers and tables are correct.

Author Response

We thank Reviewer#3 for their kind and encouraging comments. We are very grateful for the quality of this review and for how fast we have received it. Thank you very much.

Round 2

Reviewer 2 Report

Previous review comments are mostly addressed and taken into account, and the paper has improved the clarity of the exposition, structure and presentation of results.